# Relationship between urban morphology and land surface temperature—A case study of Nanjing City

**Shusheng Yin**[1,2]*, **Jiatong Liu**[1], **Zenglin Han**[1,2]

**1** School of Geography, Liaoning Normal University, Dalian, Liaoning, China, **2** Center for Studies of Marine Economy and Sustainable Development, Liaoning Normal University, Dalian, Liaoning, China

\* yinshusheng01@gmail.com

**Data Availability Statement:** Remote sensing imagery was acquired from USGS(http://glovis.usgs.gov). POI and building data were taken from the Baidu Map(http://ditu.amap.com). Land Use/Cover Change (LUCC) was acquired from(http://

## Abstract

This study investigated the relationship between urban form and land surface temperature (LST) using the Multi-access Geographically Weighted Regression (MGWR) model. A case study on Nanjing City was conducted using building data, point-of-interest (POI) data, land use data, remote sensing data, and elevation data. The results show that the MGWR model can reveal the influence of altitude, urban green space, road, building height (BH), building density (BD) and POI on LST, with a superior fitting effect over the geographically weighted regression model. LST in Nanjing exhibits a significant spatial differentiation, and the distribution of LST hotspots is spatially consistent with the level of urban construction. In terms of the two-dimensional landscape pattern, LST decreases with altitude and increases with POI. In terms of the three-dimensional structure, building height has a positive correlation with LST. POI, urban roads, and urban buildings positively affect LST, while urban green space and altitude negatively affect LST. The results of this study were verified against existing findings. The LST of areas with high-rise and super high-rise buildings is lower than that of areas with mid-rise building, which can be attributed to the large number of shadow areas formed by high-rise and super high-rise buildings. A similar phenomenon was also observed between areas with medium- and high-density buildings. These findings provide a reference for urban architecture planning and can help to develop urban heat island adaptation strategies based on local conditions.

## Introduction

With the continuous expansion and increase of the scale and number of cities in China, the boundary between urban and rural areas is becoming increasingly blurry. The urbanization rate of China's permanent population reached 60.60% in 2019(http://data.stats.gov.cn).The continuous expansion of the scale of related industrial activities [1, 2] is driving economic growth, improving the employment environment, and increasing the income of residents [3–5]. However, this expansion has also led to negative effects on the quality of human settlements, social and economic development [6–9]. In particular, the urban heat island effect,

www.resdc.cn/). DEM was downloaded on the China Academy of Sciences website(http://www.gscloud.cn) Administrative boundary was downloaded on National Platform for Common Geospatial Information Services(https://www.tianditu.gov.cn/), part of the boundary is based on OpenStreetMap(https://www.openstreetmap.org/).

**Funding:** This material is partially based upon work supported by the National Science Foundation under Grant NO. 41976206. Any opinions, findings, and conclusions or recommendations expressed in this material are those of the author and do not necessarily reflect the views of the National Science Foundation.

**Competing interests:** The authors have declared that no competing interests exist.

especially urban heat waves associated with global warming, can increase the vulnerability of populations to various health issues, such as heatstroke, and even death [10]. Therefore, improving the urban thermal environment has become the focus of relevant scholars and institutions [11]. To measure the quality of the urban thermal environment and mitigate the negative effects, it is necessary to analyze the spatiotemporal pattern of the thermal environment and its influencing factors. For this purpose, various indicators have been developed. Among them, Land Surface Temperature (LST), a basic parameter in the fields of meteorology and ecological changes [12–14], has become an important indicator [15–17], facilitating detailed analyses [18, 19].

The relationship between urban landscape patterns, such as land use type and blue-green space [20], and surface temperature has received extensive attention worldwide. At present, researchers in the field of LST studies are focusing on the following aspects: (1) The impact of the two-dimensional urban landscape pattern on LST, analyzing the impact of different land use types on LST from the perspective of land use type and land use type transformation [1, 21]; (2) Statistical analysis of the mathematical relationship between surface factors and LST using indicators, such as normalized difference vegetation index, normalized difference moisture index, normalized difference built-up index, and building density (BD) [3, 22, 23]; (3) Correlation between the three-dimensional structure and LST based on indicators, such as building height (BH), floor area ratio (FAR), and sky view factor (SVF) [24–26]. It is worth noting that the process of urbanization involves a contradiction between population concentration and limited supply of construction land, which further leads to the rapid expansion of cities in the two-dimensional direction and the continuous increase of BH [27, 28]. However, most studies investigated the relationship between surface factors and LST through simple or single regression analysis, ignoring the spatial autocorrelation and spatial information between the two. In addition, research on the relationship between urban form and LST mostly focus on a single factor of urban form (two-dimensional or three-dimensional), especially the two-dimensional form. For mega cities famous for their ecology (Gardens) (with a permanent resident population of over 8.5 million), such as Nanjing, the lack of multi-factor analysis combined with the two-dimensional landscape pattern and three-dimensional structural form makes it difficult for existing studies to further explore the impact of urban spatial differences on urban thermal environment. In addition, the urban spatial planning system has undergone major changes, and the three-dimensional urban structure is undergoing significant changes [29, 30]. Therefore, the existing approach of analyzing the urban thermal environment needs to be clarified and the two-dimensional landscape and three-dimensional space of urban landscapes need to be optimized.

This study aims to comprehensively explore the relationship between urban form (two-dimensional landscape pattern and three-dimensional structural form) and LST. For this purpose, the Multi-scale Geographically Weighted Regression (MGWR) model, a mono-window algorithm, was used to quantitatively analyze the spatial pattern of related surface factors in each local climate zone of Nanjing on LST. With this approach, this study addresses the lack of multi-factor analysis to a certain extent. The findings will provide support for regulating the urban thermal environment and improving the quality of urban human settlements.

## Study area and data sources

### Overview of the study area

Nanjing is located at 31˚14'N-32˚36'N, 118˚22'E-119˚14'E. It is a low altitude area with a humid subtropical climate (Cfa). According to meteorological records, On July 28, 2018, extreme high temperature weather occurred, which was 37.20˚C. It has 11 districts with a total area of 6587 km2 and a built-up area of 817 km2. As of 2019, permanent residents account for a population

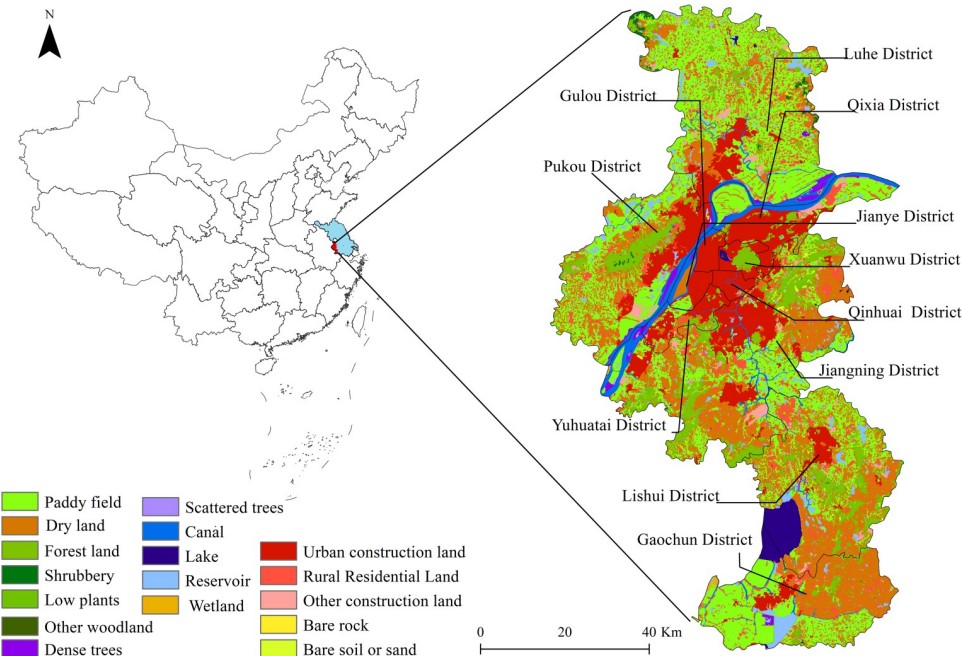

**Fig 1. Location of the study area.** The location of Nanjing in China and Jiangsu. examined in this paper, in Jiangsu Province (blue), in Nanjing City (red), and on the right side is the District of Nanjing city. This map is drawn by the authors. This map was created using ArcGIS ® software by Esri, ©National Platform for Common Geospatial Information Services, ©USGS, ©Resource and Environment Science and Data Center, ©Baidu, ©OpenStreetMap and contributors, Creative Commons Attribution 4.0(CC BY 4.0).

of 8.5 million and an urbanization rate of 83.2%. The increasing concentration of population and amount of industrial activities are exacerbating the urban heat island effect (Fig 1).

## Data source and processing

In this study, building data, POI data, land use data, remote sensing data and elevation data were selected. The data sources and descriptions are shown in Table 1. BH and BD reflect the concentration of buildings in the vertical and horizontal directions, respectively. To a certain extent, the two have the greatest impact on the three-dimensional structure of the city, and the degree of LST has the most direct effect [16, 31, 32]. According to the 2019 Unified Standards for Civil Building Design, BH is divided into 3 types: low-rise civil buildings (≤27 m), high-rise civil buildings (27–100 m), and super high-rise buildings (>100 m) [31, 33]. Considering

**Table 1. Data source and description.**

| Data | Time | Data interpretation | Data sources |
|---|---|---|---|
| Building data | 2019 | Vector data | http://ditu.amap.com |
| POI | 2019 | Point data | http://ditu.amap.com |
| Land Use/Cover Change (LUCC) | 2018 | Raster data | http://www.resdc.cn/ |
| | | Landsat 8-OLI (Resolution 30 m) | |
| Remote sensing | 2019-08-12 | Landsat 8-OLI (Resolution 30 m) | glovis.usgs.gov |
| DEM | 2009 | SRTM DEM (Resolution 30 m) | http://www.gscloud.cn/ |
| Administrative boundary | 2019 | Vector data | https://www.tianditu.gov.cn/ |
| | | | https://www.openstreetmap.org/ |

the findings of existing research and the status quo of the research area, with 20% and 40% as the boundary, BD was divided into three types: sparse, open and compact [34]. POI data were obtained through Baidu Map API, and the data were cleaned and selected for nuclear density analysis. LST data were obtained through inversion using ENVI5.3 software with Landsat 8 products, which were geometrically corrected [35]. Urban green space and aspect were obtained from digital elevation data using the ArcGIS 10.7 surface analysis function.

## Methods and factor selection

### Mono-window algorithm

LST is one of the important parameters in the study of surface energy balance. The commonly used methods of remote sensing inversion of LST mainly include the radiative transfer equation method, single window algorithm, single channel algorithm, and split window algorithm. Qin et al. [27] analyzed atmospheric water vapor content using the mono-window algorithm and found a significant negative correlation between atmospheric transmittance and the inversion error of LST [36]. Nanjing has many water bodies, and summer is mostly hot and humid. As the area features low atmospheric permeability and high accuracy of LST inversion, Landsat TM 6 band and Landsat 8 TIRS 10 were selected, and a mono-window algorithm was used to invert LST. The inversion formula can be expressed as follows:

$$T_S = \{a_6(1 - C_6 - D_6) + [b_6(1 - C_6 - D_6) + C_6 + D_6]T_6 - D_6 T_a\}/C_6 \tag{1}$$

$$C_6 = \varepsilon_6 \tau_6 \tag{2}$$

$$D_6 = (1 - \varepsilon_6)[1 + (1 - \varepsilon_6)\tau_6] \tag{3}$$

where $T_S$ is surface inversion temperature (K); $a_6$ and $b_6$ are constants; $T_6$ is luminance temperature (K); $T_a$ is the average temperature of the atmosphere (K); and $C_6$ and $D_6$ are intermediate variables, which can be obtained from $\varepsilon_6$ (surface specific emissivity) and $\tau_6$ (atmospheric transmittance in the thermal infrared band).

### MGWR model

Compared with the classic geographically weighted regression (GWR) model, the kernel function and bandwidth selection of MGWR continue the selection criteria in the classic GWR, but the MGWR model adds spatially stable variables, and each regression coefficient $\beta_{bwj}$ is based on local regression. Moreover, each bandwidth is different. The calculation formula is as follows:

$$yi = \sum_{j=1}^{k} \beta_{bwj}(u_i, v_i)x_{ij} + \varepsilon_i \tag{4}$$

where $bwj$ is the bandwidth of the regression coefficient of the *j-th* variable, $(u_i, v_i)$ represents the coordinates of the *i-th* local point in geographic space, $x_{ij}$ is the influencing factor, and $\varepsilon_i$ is the random error term.

### Selection of impact factors

According to the principle of surface heat radiation and thermodynamic characteristics, LST is affected by thermal channels and near-surface gas LST. In low-altitude areas such as Nanjing, changes in surface radiation caused by topography, impervious surfaces, human Activity significantly affect the regional surface radiation. Referring to previous studies [1, 37–40],

**Table 2. Descriptions of major explanatory variables.**

| Variable name | Unit | Description |
|---|---|---|
| Intercept | ˚C | Model intercept term, inversion of location factors. |
| Altitude | m | Altitude of Nanjing. |
| Urban Green Space | m2 | Urban green space and park distribution in Nanjing. |
| Road | Classification | Distribution of roads of grade three and above. |
| Building Height (BH) | Classification | 1 is a low-rise building, 2 is a high-rise building, and 3 is a super high-rise building. |
| Building Density (BD) | Classification | 1 is sparse, 2 is open building, 3 is compact building |
| POI | - | Spatial distribution of industrial activities. |

combined with the actual situation in the study area, and considering issues such as data availability, this study selected the factors listed in Tables 2 and 3.

## Results and analysis

### Spatial pattern characteristics of LST

Using Landsat 8 images, the overall distribution of LST was inversed according to the single window algorithm (Fig 2). The highest LST of Nanjing was 37.121˚C, the lowest was 19.923˚C, and the average was 28.525˚C. According to an analysis of hot spots, excluding outliers and the spatial distribution of LST As shown in Fig 3.

Overall, hotspots in Nanjing are concentrated mainly in Xuanwu District, Qixia District, Pukou District, Lishui District, Jiangning District, and Gaochun District. The highest value of LST was 37.12˚C in Qixia District, over an area located near the Lingang Industrial Concentration Zone in Qixia District and Nanjing Economic and Technological Development Zone. The area along the river between the two zones is largely covered by industrial land and residential land, with little vegetation. At the same time, the lowest value of LST in Nanjing was also observed in Qixia District, specifically in the Qixia Mountain. The area lies 268 m above sea level. The land use type is woodland, and the vegetation coverage is relatively high, with less interference from human activities. The distribution characteristics of LST hotspots in

**Table 3. Statistics of LST in each district.**

| District | Minimum Temperature | Maximum Temperature | Cold spot | Proportion | Hot spot | Proportion |
|---|---|---|---|---|---|---|
| Xuanwu | 22.978 | 31.570 | 413*** | 0.074 | 2172*** | 0.391 |
| Yuhuatai | 22.924 | 34.608 | 98*** | 0.030 | 331*** | 0.103 |
| Qinhuai | 24.506 | 31.744 | 24* | 0.006 | Not significant | Not significant |
| Gulou | 22.856 | 30.141 | 39*** | 0.008 | Not significant | Not significant |
| Jianye | 22.887 | 33.677 | 169*** | 0.047 | Not significant | Not significant |
| Qixia | 19.923 | 37.121 | 61*** | 0.030 | 838** | 0.410 |
| Pukou | 22.845 | 35.846 | 481*** | 0.106 | 2771** | 0.609 |
| Luhe | 20.666 | 33.437 | 927*** | 0.147 | 3452 | 0.550 |
| Lishui | 20.204 | 33.871 | 694*** | 0.131 | 3323** | 0.659 |
| Jiangning | 21.724 | 36.274 | 761*** | 0.163 | 2056*** | 0.441 |
| Gaochun | 21.104 | 35.072 | 417*** | 0.119 | 1681*** | 0.48 |

***indicates a confidence level of 0.99

** indicates a confidence level of 0.95

* indicates a confidence level of 0.90.

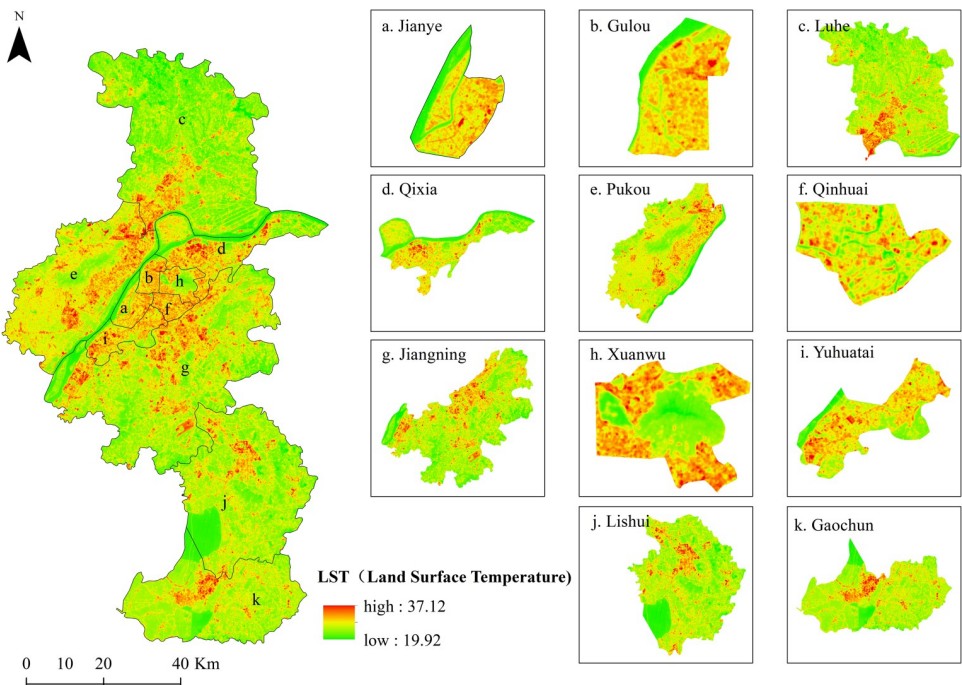

**Fig 2. Spatial distribution of LST in Nanjing.** The Land surface temperature (LST)of Nanjing city and its districts, (a) Jianye; (b) Gulou; (c) Luhe; (d) Qixia; (e) Pukou; (f) Qinhuai; (g) Jiangning; (h) Xuanwu; (i) Yuhuatai; (j)Lishui;(k) Gaochun. This map is drawn by the authors. This map was created using ArcGIS Ⓡ software by Esri, ©National Platform for Common Geospatial Information Services, ©OpenStreetMap and contributors, Creative Commons Attribution 4.0(CC BY 4.0).

other areas are related to the cluster distribution of urban land, rural residential areas and other construction land, and the scattered distribution of waters and forests and grasslands, and are strongly affected by human activities.

## Distribution of impact factors

After cleaning, calibrating, deleting anomalies, and other processing of building data, a total of 164,581 building areas were obtained (Fig 4A). Low-rise, high-rise, and super high-rise buildings accounted for 64.20%, 32.84%, and 2.96%, respectively. Low-, medium-, and high-density buildings accounted for 57.89%, 26.60%, and 15.50%, respectively. The proportion of building area in each district is as follows: Qinhuai District (29.23%) > Xuanwu District (15.17%) > Yuhuatai District (11.16%) > Jianye District (9.91%) > Jiangning District (9.91%) > Pukou District (9.65%) > Qixia District (8.75%) > Luhe District (4.02%) > Lishui District (1.751%) > Gaochun District (0.420%) > Gulou District (0.03%).

Regarding land occupied by buildings in terms of BH (Table 4), Gaochun District has fewer building areas (blocks) at 692 blocks, mainly comprising low-rise buildings with a small number of high-rise buildings. Gulou District has a very high proportion of super high-rise buildings, with a very low proportion of low-rise buildings and a moderate proportion of high-rise civil buildings. Low-rise buildings are mainly distributed in Jiangning District, Yuhuatai District, Qinhuai District, Pukou District, and Xuanwu District. Lishui District and Qixia District mainly comprise low-rise and high-rise buildings, with a relatively low proportion of high-rise buildings. Jianye District is dominated by low-rise buildings, with high-rise buildings accounting for 38.24%, and the proportion of super high-rise buildings is relatively low. The main

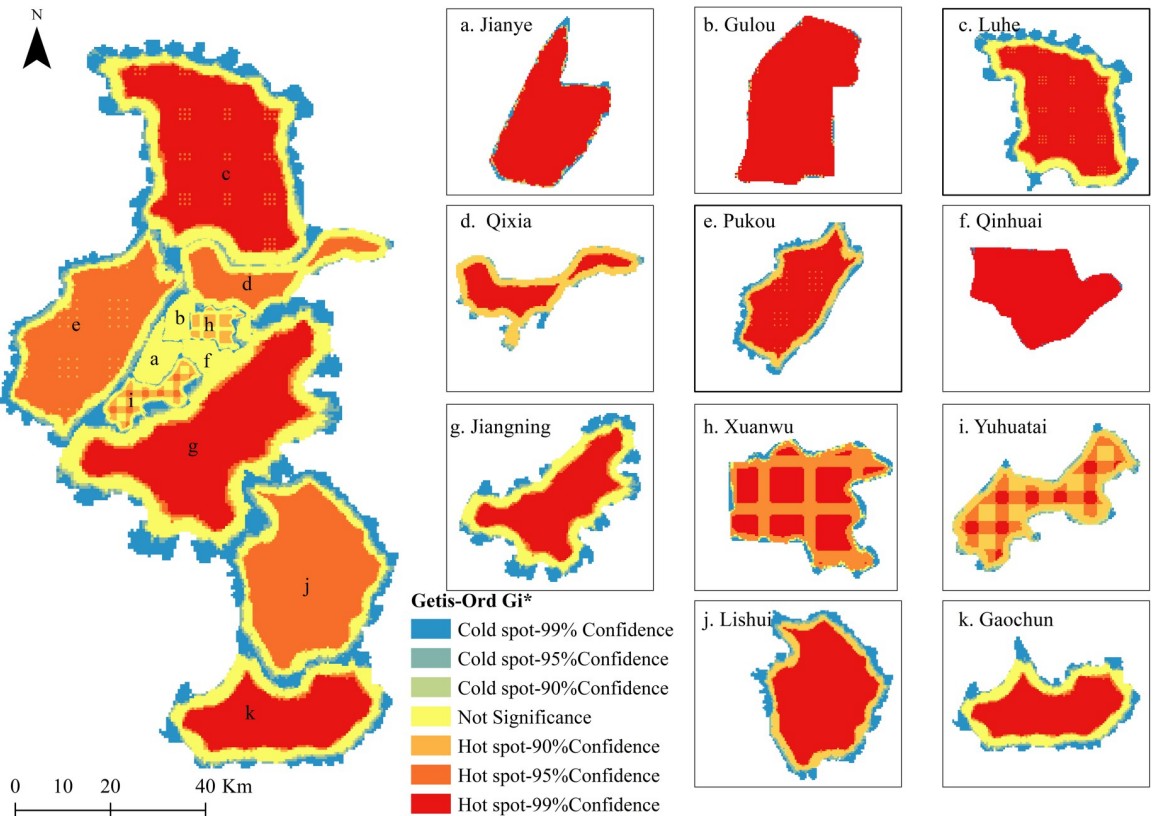

**Fig 3. Hot spot analysis of LST.** Hot spot analysis of Land surface temperature (LST). (a) Jianye; (b) Gulou; (c) Luhe; (d) Qixia; (e) Pukou; (f) Qinhuai; (g) Jiangning; (h) Xuanwu; (i) Yuhuatai; (j) Lishui; (k) Gaochun. This map is drawn by the authors. This map was created using ArcGIS ® software by Esri,©National Platform for Common Geospatial Information Services, ©OpenStreetMap and contributors, Creative Commons Attribution 4.0(CC BY 4.0).

building types in Luhe District are low-rise and high-rise buildings, with a low distribution of super high-rise buildings.

Regarding the proportion of land occupied by buildings in terms of BD (Table 4), Gaochun District is dominated by low-density buildings, with a very low proportion of medium- and high-density buildings. The proportion of buildings is relatively low in Gulou District, with high-density buildings accounting for 89.09%, and low- and medium-density buildings accounting for 10.91%. Pukou District and Xuanwu District are dominated by low-density buildings, with no medium-density buildings; Yuhuatai District and Qinhuai District mainly include low-density buildings, with some moderate and high-density buildings; Lishui District and Jiangning District mainly include low-density buildings and medium-density buildings. Jianye District has a relatively high proportion of low-density buildings, and a moderate proportion of medium- and high-density buildings. Qixia District is dominated by low- and medium-density buildings. Moreover, the proportions of the two are similar, and the distribution of high-density buildings is relatively small. Luhe District has a relatively high proportion of medium-density buildings, and moderate proportions of low- and high-density buildings.

POI represents the distribution of urban physical facilities. After data crawling and cleaning, a total of 238423 POI were obtained, with a bandwidth of 0.6 km. The core density of POI in Nanjing was 2.18–996.91/km$^2$, showing a spatial distribution of "low around and high in the center" (Fig 4B). The distribution of POI and nuclear density in each region are shown in Table 5. The overall number of POI in Jiangning District, Pukou District, and Luhe District

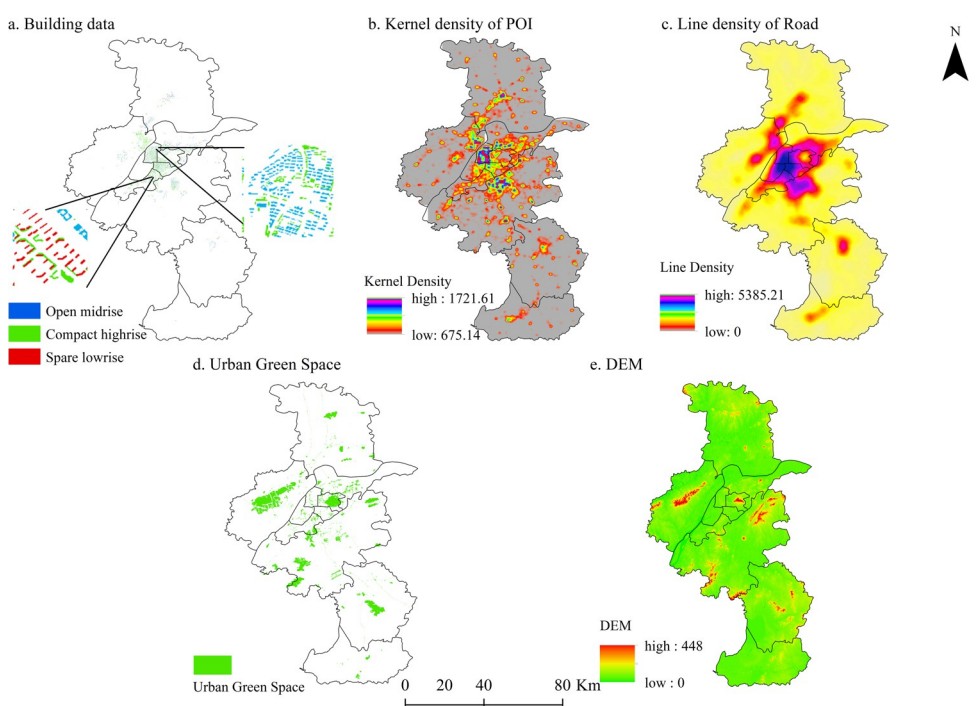

**Fig 4. Spatial distribution of impact factors.** Spatial distribution of impact factors. (a) Building date, open midrise (blue), Compare highrise (green), Spare lowrise (red),data source (http://ditu.amap.com); https://www.openstreetmap. org; (b) Kernel density of POI, data source (http://ditu.amap.com); (c) Line density of Road, data source (http://ditu. amap.com); (d) Urban Green Space(green), data source (http://www.resdc.cn/); (e) DEM, data source (http://www. gscloud.cn/). This map is drawn by the authors. This map was created using ArcGIS ® software by Esri,©National Platform for Common Geospatial Information Services, ©Resource and Environment Science and Data Center, ©Baidu, ©OpenStreetMap and contributors, Creative Commons Attribution 4.0(CC BY 4.0).

was higher than that in other districts, and the overall number of POI was the lowest in Gaochun district. With an extremely low value, the regional nuclear density was the lowest in Yuhuatai District, whereas it was the highest in Qinhuai District. The spatial clustering distribution of POI in each region exhibited different trends. Qixia District and Luhe District corresponded to the distribution of "upper low high", with percentages of points between the lowest

**Table 4. Statistics of buildings in each district.**

| District | Building area (piece) | Proportion of various building heights (BH) | | | Proportion of building density (BD) | | |
|---|---|---|---|---|---|---|---|
| | | Low-rise building | High-rise buildings | Super high-rise buildings | Spare buildings | Open buildings | Compact buildings |
| Gaochun | 692 | 96.53% | 3.47% | 0.00% | 96.24% | 3.03% | 0.72% |
| Lishui | 2881 | 50.57% | 46.27% | 3.16% | 47.66% | 33.32% | 19.02 |
| Jiangning | 16312 | 62.31% | 35.53% | 2.16% | 49.66% | 35.78% | 14.57% |
| Yuhuatai | 18362 | 65.60% | 31.19% | 3.21% | 60.11% | 22.06% | 17.84% |
| Jianye | 16317 | 55.28% | 38.24% | 6.48% | 50.59% | 24.03% | 25.39% |
| Qinhuai | 48104 | 67.48% | 30.23% | 2.29% | 62.15% | 22.19% | 15.66% |
| Gulou | 55 | 7.27% | 21.82% | 70.91% | 5.46% | 5.45% | 89.09% |
| Xuanwu | 24962 | 68.41% | 30.30% | 1.29% | 61.57% | 26.22% | 12.21% |
| Qixia | 14398 | 47.88% | 46.07% | 6.06% | 41.14% | 40.66% | 18.20% |
| Pukou | 15884 | 76.11% | 21.08% | 2.81% | 72.07% | 18.30% | 9.64% |
| Luhe | 6614 | 38.25% | 52.60% | 9.15% | 30.72% | 44.80% | 24.48% |

**Table 5. POI statistics of each district.**

| District | POI (piece) | Proportion (%) | Density (pcs/km$^2$) | Standard deviation |
|---|---|---|---|---|
| Gaochun | 10517 | 4.411 | 2.852–727.334 | 65.092 |
| Lishui | 14788 | 6.202 | 2.938–749.145 | 62.990 |
| Jiangning | 59429 | 24.926 | 3.046–776.841 | 94.372 |
| Yuhuatai | 16329 | 6.849 | 2.183–556.779 | 112.759 |
| Jianye | 10592 | 4.443 | 2.677–682.616 | 149.710 |
| Qinhuai | 15374 | 6.448 | 3.909–996.906 | 193.122 |
| Gulou | 18938 | 7.943 | 3.540–902.626 | 183.731 |
| Xuanwu | 14226 | 5.967 | 3.085–786.590 | 145.226 |
| Qixia | 23998 | 10.065 | 2.377–606.095 | 91.722 |
| Pukou | 28579 | 11.987 | 2.841–724.402 | 92.142 |
| Luhe | 25653 | 10.759 | 2.332–594.628 | 72.375 |
| Total | 238423 | 100 | | |

value and standard deviation of 61.16% and 53.49%. Jiangning District and Yuhuatai District showed a distribution trend of "low in the lower part and high in the upper part", with percentages of points between the lowest value and standard deviation of 61.66% and 52.73%, respectively. Pukou District and Jianye District showed a distribution trend of "left low and right high", with percentages of 56.915% and 56.64%, respectively, for points between the minimum value and standard deviation. Xuanwu District and Qinling District showed percentages of 56.915% and 56.64%, respectively. Huaihe District exhibited a distribution trend of "right low, left high" with percentages of 51.86% and 43.01%. Gaochun District and Lishui District showed distribution trends of "low edge, high middle", with percentages of 52.66% and 51.64%. The overall distribution of POI nuclear density in Gulou District was uniform.

After checking the topology of road network data, they were calibrated based on Google Earth Pro2019. Referring to existing research and the actual situation of the research area, highways, urban traffic arterial roads, and urban branch roads were reserved (Fig 4C), setting a 1 km × 1 km grid, and the road network density (road network density = total road length km/grid area km$^2$) was calculated. The results are shown in Table 6, for which abnormal values were removed. The higher the road network density, the stronger the regional road accessibility, promoting travel among urban residents. The higher the connectivity, the more convenient the regional traffic; conversely, the lower the density of the road network, the less the

**Table 6. Road statistics of each district.**

| District | Road network density (km/km$^2$) | Standard deviation (km/km$^2$) |
|---|---|---|
| Gaochun | 0.011–15.645 | 2.933 |
| Lishui | 0.009–15.516 | 3.436 |
| Jiangning | 0.004–15.573 | 3.853 |
| Yuhuatai | 0.037–18.111 | 5.377 |
| Jianye | 0.068–16.816 | 6.174 |
| Qinhuai | 1.109–14.263 | 4.910 |
| Gulou | 0.143–15.805 | 5.647 |
| Xuanwu | 0.110–14.534 | 5.163 |
| Qixia | 0.009–12.112 | 4.431 |
| Pukou | 0.004–14.918 | 3.199 |
| Luhe | 0.002–12.878 | 3.371 |

distribution of arterial roads and highways in the area, which greatly reduces road accessibility and decreases the convenience of residents to travel. Combining Figs 1 and 2, the road network in Nanjing shows uneven development. Areas with high levels of road network density are mainly distributed in the central area, including the districts of Jianye, Gulou, and Yuhuatai, which may be clustered with buildings. It is related to the dense distribution of entertainment venues such as cultural centers. Areas with low levels of road network density are mainly distributed in the south, northeast, and northwest regions, such as the districts of Gaochun, Qixia, Luhe, and Pukou, which may have large areas of water and forests. The distribution of grassland and farmland is related to the scattered distribution of rural settlements.

In terms of altitude and urban green space, Nanjing is in a low altitude area, dominated by lacustrine plains and valley bottoms, with a small amount (39.2% of the total area) of undulating mountains and hills, plains, depressions, rivers, and lakes. The Ningzhen Mountains and Jiangbei Laoshan straddles the central part of the city, and the south is a geomorphologically complex area composed of topographical units, such as low mountains, valley plains, and rivers. As mountainous areas, the highest altitudes of Xuanwu, Pukou, and Jiangning districts are higher than those of other regions (Fig 4E). The density of urban green space (total area of urban gardens and green space/total area of urban land) is shown in Fig 4D. The urban green space in Nanjing shows a widely variable spatial distribution, mainly concentrated in Xuanwu Lake, Zijin Mountain, and ancient city walls in the city center. Overall, in addition to urban green spaces near the "Central Park", they are concentrated in the Riverside Scenic Belt, Qinhuai River Scenic Belt, and Pukou Central Park.

### Relationship between LST and impact factors

To quantitatively analyze the spatial distribution characteristics of LST and its factors in Nanjing, altitude, urban green space, slope, aspect, BD, BH, and POI were analyzed. A correlation test of the variables was conducted, and the results showed that the *Moran's* between slope and aspect is less than 0.2, with a small spatial correlation. As this index failed the *Moran's* test, it was not included in the global variable. The remaining indexes all showed values greater than 0.7, with some at the 1% level, showing significant spatial positive correlation. Accordingly, they were included in the local variables for calculation.

Table 7 shows that the goodness of fit ($R^2$) of MGWR is slightly higher than that of the precision GWR model, and the value of the corrected Akaike information criterion (AICc) is also lower than that of the classic GWR model. Therefore, MGWR can be assessed to have higher performance than the classic GWR. Comparing the residual sum of squares, the value of MGWR was also smaller than that of GWR. Moreover, MGWR requires fewer parameters to obtain a regression result closer to the true value.

The spatial distribution of each influencing factor is shown in Fig 4. The MGWR analysis results show that the spatial distribution of each influencing factor and LST have significant

**Table 7. Model index of GWR and MGWR.**

| Model | MGWR | GWR |
|---|---|---|
| $R^2$ | 0.928 | 0.883 |
| $AIC_C$ | 1303 | 1186 |
| Adj. $R^2$ | 0.902 | 0.875 |
| Residual sum of squares | 1323 | 1977 |
| Sig. | 0.000* | 0.000* |

* Indicates that the test passed at the 5% significance level.

similarities and differences. LST decreased with increasing altitude, urban green space, and BH, whereas it increased with increasing values of BD and POI. The goodness of fit between POI and LST was the highest, reaching 0.96, followed by that between BD and LST (0.95) and between BH and LST (0.94). The values of goodness of fit between altitude, urban green space and LST were good (0.73, 0.61, respectively), and that between aspect and LST was average (0.37). These results further show that the natural environment is the basic factor affecting the spatial distribution of LST, and changes in the surface environment caused by human activities have a particularly significant impact on urban LST. Industrial facilities and business districts, such as Xinjiekou business district in Qinhuai District, Taipingmen business district in Xuanwu District, and Shuiximen business district in Jianye District, are concentrated in Xuanwu District, Qinhuai District, Gulou District, Jianye District, and other downtown areas, which also account for the main low-rise buildings and super-high buildings in Nanjing. At the same time, a large number of buildings are concentrated in the central area, which further increases the flow of people. The concentrated POI distribution has a significant positive correlation with the spatial distribution of LST.

The statistical description of each coefficient of MGWR is shown in Table 5. The *Intercept* represents the positive influence of location factors on LST. The value of *Intercept* was between -0.49 and 1.46, the average value was 0.485, and the standard deviation was 0.975, indicating that under the same natural conditions, the influencing factors would change the LST of Nanjing by -0.49–1.46˚C, with an average change of 0.485˚C. The influence of location factors on LST widely varies.

Urban roads have a significant positive impact on LST (Table 8). Herein, roads in Nanjing are divided into three levels according to the classification standards of urban roads: *Arterial Road*, *Secondary Road*, and *Access Road*. *Arterial Road* ranges from -0.42 to 1.41, with an average value of 0.495, *Secondary Road* ranges from -0.34 to 1.53, with an average value of 0.595, and *Access Road* ranges from 0.76 to 1.97, with an average value of 1.365. These results show that in urban roads, branch roads have a greater impact on LST. In other words, under the circumstance that the influence of other factors remains unchanged, the LST around a branch road (*Access Road*) would be approximately 1.365˚C higher than that in the surrounding areas. Therefore, the density of urban roads will also affect LST.

Referring to the literature [41], quantitative indicators of local climate zones, and the actual situation of Nanjing, the study area was divided into low-density low-rise building areas,

**Table 8. Statistical description of MGWR coefficients.**

| Variable | Min | Max | Median | Std | Mean |
|---|---|---|---|---|---|
| Intercept | -0.49 | 1.46 | 0.485 | 0.975 | 0.485 |
| Arterial Road | -0.42 | 1.41 | 0.495 | 0.915 | 0.495 |
| Secondary Road | -0.34 | 1.53 | 0.595 | 0.935 | 0.595 |
| Access Road | 0.76 | 1.97 | 1.365 | 0.605 | 1.365 |
| BuildingHeight.1 | -0.14 | 0.21 | 0.035 | 0.175 | 0.035 |
| BuildingHeight.2 | 0.45 | 3.01 | 1.73 | 1.28 | 1.73 |
| BuildingHeight.3 | -0.6 | 3.76 | 1.58 | 2.18 | 1.58 |
| Urban Green Space | -1.42 | 0.26 | -0.58 | 0.84 | -0.58 |
| Compact Highrise | -0.43 | 0.67 | 0.14 | 0.55 | 0.12 |
| Open Midrise | -0.52 | 3.68 | 1.55 | 2.1 | 1.58 |
| Lightweight Rise | 0.29 | 1.2 | 0.741 | 0.455 | 0.745 |
| POI | -0.72 | 1.6 | 0.47 | 1.16 | 0.44 |
| DEM | -2.5 | -0.25 | -1.381 | 1.125 | -1.375 |

medium-density high-rise building, areas and high-density super high-rise building area (Fig 4E), and the relationships between BH, BD, and LST were analyzed. Low-rise building areas (*BuildingHeight.1*) showed LST values ranging from -0.14 to 0.21, with an average value of 0.035. High-rise building areas (*BuildingHeight.2)* showed LST values ranging from 0.45 to 3.01, with an average value of 1.73. Super high-rise building areas (*BuildingHeight.3*) showed LST values ranging from -0.6–3.76, with an average value of 1.58. This further shows that there is a significant positive correlation between BH and LST. From the perspective of the absolute value of the coefficient, the area of middle-rise buildings has the greatest impact on LST. In terms of BD, the value of the impact of high-density high-rise buildings on LST ranged from -0.43 to 0.67, with an average value of 0.12˚C. The value for medium-density high-rise building areas ranged from -0.52 to 3.68, with an average value of 1.58˚C. The value for low-density low-rise buildings ranged from 0.29 to 1.2, with an average value of 0.745˚C. The absolute value of the influence coefficient shows that high-rise buildings and medium-density mid- and high-rise buildings have the greatest influence on LST. Previous studies suggested that increases in BH will increase LST. In contrast, this study shows that increases in BH do not necessarily lead to an increase in LST. This can be attributed to increased shadow areas generated by high-rise buildings, leading to lower temperatures within a certain range [42].

Urban green space and LST in Nanjing exhibit a negative correlation, with an impact coefficient of -0.58˚C, indicating that increases in urban green space will decrease LST. Under the condition that other factors remain unchanged, the LST of urban green space is lower than that of surrounding areas by 0.58˚C. In contrast, POI has a positive correlation with LST. The higher the intensity of human activities, the higher the LST, but its impact coefficient is 0.44, which is moderate. The DEM coefficient ranged from -2.5 to -0.25. This implies that the LST of areas at higher altitudes is lower than that of areas at lower altitudes by 1.375˚C on average.

## Conclusion

In this study, the MGWR model was used to determine the relationship between urban form and surface temperature for the first time. Combining POI data, building data, and remote sensing data of Nanjing, the spatial differentiation of urban form (two-dimensional landscape pattern and three-dimensional structure form) and LST and its influencing factors were analyzed. The following main conclusions can be drawn:

(1) The spatial differentiation of LST in Nanjing is significant, and the distribution of LST hotspots exhibits a distinct spatial consistency with the level of urban construction. The highest LSTs in Qixia District (37.121˚C), Jiangning District (36.274˚C), and Pukou (35.846˚C) are higher than those in other areas, and the lowest LSTs in Qixia District (19.923˚C), Lishui District (20.204˚C), and Luhe District (20.666˚C) are lower than those in other areas.

(2) Compared with classic GWR, MGWR supports the analysis of multiple influencing factors or variables at different scales and provides better fitting effect than GWR. In this study, some factors exhibited significant differences in their effects on LST. Except for the aspect and slope, the other influencing factors all showed significant spatial correlation with LST. In terms of the two-dimensional landscape pattern, the higher the altitude, the lower the LST; the higher the POI concentration, the higher the LST. In terms of the three-dimensional structure, BH and LST are positively correlated.

## Advantages and limitations

The conclusions of this research and existing research results are mutually confirmed [39, 43, 44]. In plain areas with low altitude and gentle slopes, the development of human activities is less difficult, human activities are frequent, and construction land is concentrated. Areas with

concentrated secondary and tertiary industries are more likely to have higher LST, whereas areas with mountains and agricultural land are likely to have lower LST due to several factors such as altitude and vegetation [45, 46]. It is noteworthy that, unlike the findings of previous studies, an obvious positive correlation was found between urban BH, BD, and LST. However, the LST of high-rise and super high-rise building areas was found to be lower than that of mid-rise building areas. This could be explained by the expansion of shadow areas generated by super high-rise buildings; similar phenomena were also observed between medium-density building areas and high-density building areas [42, 47].

Based on the existing research, this paper determines the impact of different human activities on the urban thermal environment, and further proves that urban green space can help alleviate the urban heat island effect [22, 48]. The process of urbanization has led to overpopulation and excessive industrial concentration, causes a change in the nature of heat exchange at the bottom, and aggravating the formation and development of the urban heat island effect, which requires the attention of urban planning agencies [49]. In addition, based on the results of this article, strategies to reduce heat stress by addressing the urban heat island effect, (e.g., control the scale of urban built-up areas, optimize urban spatial structure, increase urban green areas, alleviate urban population concentration and other measures). We should also cooperate with commercial real estate developers to control the height and density of new buildings, optimize the design of future urban parks, increase the construction of urban ventilation corridors and green spaces, and further alleviate the urban heat island effect [8].

## Advantages

This study integrates POI data, building data, urban road data sets and other data to analyze the factors that affect the urban thermal environment. First, in order to assess the human activities that may be responsible for the model described here, we compare land use data with population density data, etc., and conclude that the POI data represents the geographic information and utilization characteristics of various facilities. Secondly, for the classification of building height in Nanjing, after many field investigations and analysis of historical remote sensing images, a more reasonable density classification (limited to 40%) is finally determined. This research will have greater practical value. Third, this study uses a multi-dimensional perspective (two-dimensional and three-dimensional structure) to study the current status of Nanjing's thermal environment, and explores the current status of Nanjing's thermal environment from a more specific plane dimension and a deeper perspective. In view of the multi-dimensional perspective of this study, MGWR is used instead of GWR to explore non-stationary relationships in the modeling space [22, 44].

Multi-scale Geographically Weighted Regression (MGWR) is a recent advancement to the classic GWR model. Compared with the traditional GWR model, The MGWR model has advantages in acquiring the ability of different scales [18]. The MGWR model can effectively analyze the multi-scale relationship between the urban thermal environment and its influencing factors, and has a positive effect on urban dynamic development and urban thermal environment management.

## Limitations

The MGWR model facilitated multi-factor analysis of LST. However, due to issues such as data availability and collinearity, the application of the MGWR model has certain limitations. The urban landscape is a complex dynamic system composed of infrastructure, human activities, and social connections. Changes in urban surface temperature need to be observed from a more micro perspective [8, 50]. Urban ground monitoring data have not been fully disclosed,

which limits the study. In addition, street view data were used in the study of the urban thermal environment, and the number of street scenes in this area requires further investigation [51]. In the future, the interaction between different influencing factors should be considered, and the influencing factors of LST should be analyzed in more detail to provide a more comprehensive perspective for urban or regional environmental governance and planning.

## Acknowledgments

The authors thank the various members of the research program for their professional contributions and engagement. We are thankful to all the experts' contributions in the building of the model and the formulation of the strategies in this study.

## Author Contributions

**Conceptualization:** Shusheng Yin.

**Data curation:** Shusheng Yin, Jiatong Liu.

**Formal analysis:** Jiatong Liu, Zenglin Han.

**Funding acquisition:** Zenglin Han.

**Investigation:** Zenglin Han.

**Methodology:** Shusheng Yin, Jiatong Liu.

**Resources:** Jiatong Liu.

**Software:** Jiatong Liu.

**Visualization:** Shusheng Yin.

**Writing – original draft:** Shusheng Yin, Zenglin Han.

**Writing – review & editing:** Zenglin Han.

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
