## [Decision Letter · Decision Letter 0]

6 Mar 2021

PONE-D-21-03097

Relationship between urban morphology and land surface temperature – a case study of Nanjing City

PLOS ONE

Dear Dr. shusheng,

Thank you for submitting your manuscript to PLOS ONE. After careful consideration, we feel that it has merit but does not fully meet PLOS ONE’s publication criteria as it currently stands. Therefore, we invite you to submit a revised version of the manuscript that addresses the points raised during the review process.

We look forward to receiving your revised manuscript.

Kind regards,

Jun Yang

Academic Editor

PLOS ONE

Additional Editor Comments:

Reviewer 1

This paper is overall interesting. However, the major revisions are required in terms of English, references, logic and structure.

Please see the detailed comments in the attachment. My pleasure to see your revised verison.

Reviewer 2

Summary: The paper presents an application of MGWR to Landsat-derived LST across Nanjing. This is potentially useful to other researchers as an application case study that can inform future work on similar problems. I recommend "major" revisions primarily b/c I'm asking for a bit of additional statistical analysis. But I am confident that the authors can address my comments.

Section 3.2: Since the purpose of this paper is to compare MGWR to GWR, it would be useful if the authors could expand this section to provide a bit more context. Has MGWR been applied to similar problems elsewhere? What is the theoretical motivation for applying MGWR to the challenge of LST prediction in an urban environment?

Section 4.1: The first long paragraph is mostly unnecessary b/c the results are summarized in a table. I suggest that the authors remove this paragraph in favor of a short paragraph that highlights any particularly interesting features of these results. A similar comment applies to the Section 4.2 text that presents the numbers preported in Tables 4 and 5.

Section 4.3: Is MGWR better than GWR by a statistically significant amount? A test of statistical significance could be added to "Table 4" of this section (which I think is actually Table 7 in the manuscript).

Section 4.3: Are the R2 statistics provided here for data that were used in fitting the MGWR and GWR, or for holdout data? While AIC is useful, it would be better to see how the models compare in out-of-sample prediction. Also, why is R2 and adjR2 the only metric used? What about MSE, or some other indicator of magnitude of error?

Section 5: It is unusual to have a "Conclusions" section between Results and Discussion. Perhaps this could be restructured to be a Section 4.4 "Summary" of results?

p. 26, top line: there's an incomplete sentence here.

Journal Requirements:

2. We note that Figures 1-4 in your submission contain map images which may be copyrighted. All PLOS content is published under the Creative Commons Attribution License (CC BY 4.0), which means that the manuscript, images, and Supporting Information files will be freely available online, and any third party is permitted to access, download, copy, distribute, and use these materials in any way, even commercially, with proper attribution. For these reasons, we cannot publish previously copyrighted maps or satellite images created using proprietary data, such as Google software (Google Maps, Street View, and Earth). For more information, see our copyright guidelines: http://journals.plos.org/plosone/s/licenses-and-copyright.

2.1.    You may seek permission from the original copyright holder of Figures 1-4 to publish the content specifically under the CC BY 4.0 license. 

2.2.    If you are unable to obtain permission from the original copyright holder to publish these figures under the CC BY 4.0 license or if the copyright holder’s requirements are incompatible with the CC BY 4.0 license, please either i) remove the figure or ii) supply a replacement figure that complies with the CC BY 4.0 license. Please check copyright information on all replacement figures and update the figure caption with source information. If applicable, please specify in the figure caption text when a figure is similar but not identical to the original image and is therefore for illustrative purposes only.

Reviewers' comments:

Reviewer's Responses to Questions

**Comments to the Author**

1. Is the manuscript technically sound, and do the data support the conclusions?

Reviewer #1: Yes

Reviewer #2: Partly

2. Has the statistical analysis been performed appropriately and rigorously? 

Reviewer #1: Yes

Reviewer #2: Yes

3. Have the authors made all data underlying the findings in their manuscript fully available?

Reviewer #1: Yes

Reviewer #2: Yes

4. Is the manuscript presented in an intelligible fashion and written in standard English?

Reviewer #1: No

Reviewer #2: Yes

5. Review Comments to the Author

Reviewer #1: Dear authors,

This paper is overall interesting. However, the major revisions are required in terms of English, references, logic and structure.

Please see the detailed comments in the attachment. My pleasure to see your revised verison.

Reviewer #2: Summary: The paper presents an application of MGWR to Landsat-derived LST across Nanjing. This is potentially useful to other researchers as an application case study that can inform future work on similar problems. I recommend "major" revisions primarily b/c I'm asking for a bit of additional statistical analysis. But I am confident that the authors can address my comments.

Section 3.2: Since the purpose of this paper is to compare MGWR to GWR, it would be useful if the authors could expand this section to provide a bit more context. Has MGWR been applied to similar problems elsewhere? What is the theoretical motivation for applying MGWR to the challenge of LST prediction in an urban environment?

Section 4.1: The first long paragraph is mostly unnecessary b/c the results are summarized in a table. I suggest that the authors remove this paragraph in favor of a short paragraph that highlights any particularly interesting features of these results. A similar comment applies to the Section 4.2 text that presents the numbers preported in Tables 4 and 5.

Section 4.3: Is MGWR better than GWR by a statistically significant amount? A test of statistical significance could be added to "Table 4" of this section (which I think is actually Table 7 in the manuscript).

Section 4.3: Are the R2 statistics provided here for data that were used in fitting the MGWR and GWR, or for holdout data? While AIC is useful, it would be better to see how the models compare in out-of-sample prediction. Also, why is R2 and adjR2 the only metric used? What about MSE, or some other indicator of magnitude of error?

Section 5: It is unusual to have a "Conclusions" section between Results and Discussion. Perhaps this could be restructured to be a Section 4.4 "Summary" of results?

p. 26, top line: there's an incomplete sentence here.

6. PLOS authors have the option to publish the peer review history of their article (what does this mean?). If published, this will include your full peer review and any attached files.

Reviewer #1: No

Reviewer #2: No

---

## [Author Response · Author response to Decision Letter 0]

14 Sep 2021

Dear Roland Paile Bendaña

 Thank you for sending the feedback. Overall, the manuscript has been extensively revised to address the concerns raised by the editor. The specific comments and changes are listed as follows in a point-to-point manner.

Response to Editor Comments:

1. Thank you for uploading your figures as individual figure files. Before we can proceed, please still remove the images from the manuscript file, and upload an updated version of this doc. 

Response #1: We have removed the pictures in the manuscript and submitted a new manuscript.

Thanks again for the comments and valuable suggestions to improve our manuscript.

Kind regards,

Shusheng Yin.

---

## [Decision Letter · Decision Letter 1]

5 Oct 2021

PONE-D-21-03097R1Relationship between urban morphology and land surface temperature – a case study of Nanjing CityPLOS ONE

Dear Dr. shusheng,

Thank you for submitting your manuscript to PLOS ONE. After careful consideration, we feel that it has merit but does not fully meet PLOS ONE’s publication criteria as it currently stands. Therefore, we invite you to submit a revised version of the manuscript that addresses the points raised during the review process.

We look forward to receiving your revised manuscript.

Kind regards,

Jun Yang

Academic Editor

PLOS ONE

Additional Editor Comments (if provided):

After reading through authors revision, I found authors have made most revisions. However, it cannot meet the requirements.

1. For instance, several abbrevations have been used in the abstract, but it does not provide the full name.

2. Authors have inserted the solid references to support their argument in the context. However, such references have not been updated in the last Reference section. Authors have to address this problem. I will further check this in the next round of review.

3. Authors may be interested in the following references: Yang, J., Wang, Y., Xue, B., Li, Y., Xiao, X., Xia, J. C., & He, B. (2021). Contribution of urban ventilation to the thermal environment and urban energy demand: Different climate background perspectives. Science of The Total Environment, 795, 148791.

Luo, X., Yang, J., Sun, W., & He, B. (2021). Suitability of human settlements in mountainous areas from the perspective of ventilation: A case study of the main urban area of Chongqing. Journal of Cleaner Production, 310, 127467.

4. Moreover， authors are required to provide a document of 'responses to reviewers' in which authors' responses are provided point-by-point..

5. Therefore, please reply to me in the next round based on the document of 'responses to reviewers'. Both the first-round and second-round responses should be provided.

6. Section 6 should be shortened to provide key information.

Reviewers' comments:

Reviewer's Responses to Questions

**Comments to the Author**

1. If the authors have adequately addressed your comments raised in a previous round of review and you feel that this manuscript is now acceptable for publication, you may indicate that here to bypass the “Comments to the Author” section, enter your conflict of interest statement in the “Confidential to Editor” section, and submit your "Accept" recommendation.

Reviewer #1: (No Response)

Reviewer #2: All comments have been addressed

2. Is the manuscript technically sound, and do the data support the conclusions?

Reviewer #1: Partly

Reviewer #2: Yes

3. Has the statistical analysis been performed appropriately and rigorously? 

Reviewer #1: Yes

Reviewer #2: Yes

4. Have the authors made all data underlying the findings in their manuscript fully available?

Reviewer #1: Yes

Reviewer #2: (No Response)

5. Is the manuscript presented in an intelligible fashion and written in standard English?

Reviewer #1: No

Reviewer #2: Yes

6. Review Comments to the Author

Reviewer #1: After reading through authors revision, I found authors have made most revisions. However, it cannot meet the requirements.

1. For instance, several abbrevations have been used in the abstract, but it does not provide the full name.

2. Authors have inserted the solid references to support their argument in the context. However, such references have not been updated in the last Reference section. Authors have to address this problem. I will further check this in the next round of review.

3. Authors may be interested in the following references: Yang, J., Wang, Y., Xue, B., Li, Y., Xiao, X., Xia, J. C., & He, B. (2021). Contribution of urban ventilation to the thermal environment and urban energy demand: Different climate background perspectives. Science of The Total Environment, 795, 148791.

Luo, X., Yang, J., Sun, W., & He, B. (2021). Suitability of human settlements in mountainous areas from the perspective of ventilation: A case study of the main urban area of Chongqing. Journal of Cleaner Production, 310, 127467.

4. Moreover， authors are required to provide a document of 'responses to reviewers' in which authors' responses are provided point-by-point..

5. Therefore, please reply to me in the next round based on the document of 'responses to reviewers'. Both the first-round and second-round responses should be provided.

6. Section 6 should be shortened to provide key information.

Reviewer #2: I thank the authors for their responses to my queries, and I am happy to recommend the manuscript for publication.

7. PLOS authors have the option to publish the peer review history of their article (what does this mean?). If published, this will include your full peer review and any attached files.

Reviewer #1: No

Reviewer #2: No

---

## [Author Response · Author response to Decision Letter 1]

29 Oct 2021

Dear editors and reviewers:

 Thank you very much for your letter, and the referees’ reports. Based on your comment and request, we have made extensive modification on the original manuscript. Here, we attached the revised manuscript in the formats of both manuscript and editable words for your approval. A document answering every question from the referees was also summarized and enclosed.

A revised manuscript with the correction sections blue marked was attached as the supplemental material and for easy check and editing purpose. If you have any questions, please contact us without hesitation.

Comment 1:For instance, several abbrevations have been used in the abstract, but it does not provide the full name.

Answers to comment 1:

Thank you very much for referees’ reports. I seriously thought about the reviewer’s opinion and carefully answered the question. Based on your question about the full name of academic term , I checked the manuscript specifically.  For the first 

time, the explanation for the POI has appeared in the Line 11 of the abstract section .

Original(L8 - L11 ):

This study investigated the relationship between urban form and land surface temperature (LST) using the Multi-access Geographically Weighted Regression (MGWR) model. A case study on Nanjing City was conducted using building data, point-of-interest (POI) data, land use data, remote sensing data, and elevation data. 

Amendment(L8 - L11 ):

 This study investigated the relationship between urban form and land surface temperature (LST) using the Multi-access Geographically Weighted Regression (MGWR) model. A case study on Nanjing City was conducted using building data, point-of-interest (POI) data, land use data, remote sensing data, and elevation data. 

Comment 2:Authors have inserted the solid references to support their argument in the context. However, such references have not been updated in the last Reference section. Authors have to address this problem. I will further check this in the next round of review.

Answers to comment 2: 

Thank you very much for referees’ reports. I seriously thought about the reviewer's opinion and carefully answered the question. During the writing process, I have updated all the cited information in the manuscript after careful inspection. On the premise of following the journal citation rules and tracking the latest research 

in 2021, it is an honor and pleasure that some inspiration collided in my brain. I, based on the above, amended some expressions of the manuscript, in addition, and adequately proofread the references many times to ensure the rationality and scientific. Once again, I sincerely thank the referees for their careful reports and predecessors for their painstaking research.

Original(L426 - L432 ):

The conclusions of this research and existing research results are mutually confirmed (Han et al., 2016; Yan et al., 2019). In plain areas with low altitude and gentle slopes, the development of human activities is less difficult, human activities are frequent, and construction land is concentrated. Areas with concentrated secondary and tertiary industries are more likely to have higher LST, whereas areas with mountains and agricultural land are likely to have lower LST due to several factors such as altitude and vegetation. 

Amendment(L427 - L440 ):

The conclusions of this research and existing research results are mutually confirmed (Han et al., 2014; Yan et al., 2019;Yang et al.,2020). In plain areas with low altitude and gentle slopes, the development of human activities is less difficult, human activities are frequent, and construction land is concentrated. Areas with concentrated secondary and tertiary industries are more likely to have higher LST, whereas areas with mountains and agricultural land are likely to have lower LST due to several factors such as altitude and vegetation(Luo et al.,2021;Peng et al.,2021). It is noteworthy that, unlike the findings of previous studies, an obvious positive correlation was found between urban BH, BD, and LST. However, the LST of high-rise and super high-rise building areas was found to be lower than that of mid-rise building areas. This could be explained by the expansion of shadow areas generated by super high-rise buildings; similar phenomena were also observed between medium-density building areas and high-density building areas (Meng et al., 2018; Yang et al., 2021).

Originall(L528 - L732 ):

Reference

1.Alexander Buyantuyev, Jianguo Wu. Urban heat islands and landscape heterogeneity: linking spatiotemporal variations in surface temperatures to land-cover and socioeconomic patterns [J]. Landscape Ecology, 2010,25:17–33. DOI 10.1007/s10980-009-9402-4.

2.Bin Zhou, Diego Rybski, Jürgen P. Kropp. The role of city size and urban form in the surface urban heat island [J]. Science Reports, 2017,(07):1-9. DOI.10.1038/s41598-017-04242-2.

3.Blessing Bolarinwa Fabeku, Ifeoluwa Adebowale Balogun, Suleiman Abdul-Azeez Adegboyega, Orimoloye Ipoola Faleyimu. Spatio-Temporal Variability in Land Surface Temperature and Its Relationship with Vegetation Types over Ibadan, South-Western Nigeria [J]. Atmospheric and Climate Sciences, 2018,8(3):319-336.DOI:10.4236/acs.2018.83021.

4.Brunsdon C, Fontheringham A S, Charlton M E. Geographically Weighted Regression:A Method for Explor ing Spatial Nonstationarity [J]. Geographically Analysis, 1996,28(4):281-298. DOI:10.1111/j.1538-4632.1996.tb00936.x.

5.Cao Jie, Zhou Weiqi, Zheng Zhong, Ren Tian, Wang Weimin. Within-city spatial and temporal heterogeneity of air temperature and its relationship with land surface temperature [J]. Landscape and Urban Planning, 2021,206. DOI: 10.1016/j.landurbplan.2020.103979.

6.Cheng He, Liang Ma, Liguo Zhou, HaiDong Kan, Yan Zhang, WeiChun Ma, Bin Chen. Exploring the mechanisms of heat wave vulnerability at the urban scale based on the application of big data and artificial societies [J]. Environment International, 2019,127. 573-583.DOI:10.1016/j.envint.2019.01.057.

7.Dechao Chen, Xinliang Xu, Zongyao Sun, Luo Liu, Zhi Qiao, Tai Huang. Assessment of Urban Heat Risk in Mountain Environments: A Case Study of Chongqing Metropolitan Area, China[J]. Sustainability,2019,12(1). DOI:10.3390/su12010309.

8.Decheng Zhou, Jingfeng Xiao, Stefania Bonafoni, Christian Berger, Kaveh Deilami, Yuyu Zhou, Steve Frolking,R ui Yao, Zhi Qiao ,José A. Sobrino. Satellite Remote Sensing of Surface Urban Heat Islands: Progress, Challenges, and Perspectives[J]. Remote Sensing,2018,11(1). DOI:10.3390/rs11010048.

9.Feng Lei, Zhang Kaiyi, Gao Jingjing, Dai Zhixiu. Study of road surface temperature by using the land surface temperature retrieval products from meteorological satellite data [J]. Science and Technology Review, 2019,37(20):49-64.

10.Gu Guofeng, Li Qiao, Zhou Yinan. Evolution of Economic Growth Spatial Correlation Network Structure of Urban Agglomeration in Northeast China [J]. Area Research and Development, 2020,39(02):14-19.DOI:10.3969 /j. issn.1003-2363.2020.02 .003.

11.Gu Hengyu, Qin Xiaoling, Shen Tiyan. Spatial variation of migrant population's return intention and its determinants in China's prefecture and provincial level cities [J]. Geographical Research, 2019,38(08):1877-1890. DOI: 10.11821/dlyj020180473.

12.Guo, A., Yang, J., Sun, W., Xiao, X., Cecilia, J. X., Jin, C., & Li, X. Impact of urban morphology and landscape characteristics on spatiotemporal heterogeneity of land surface temperature[J]. Sustainable Cities and Society, 2020,63: 102443. DOI: 10.1016/j.scs.2020.102443.

13.Han G F, Ye L, Sun Z W. Influence of aspect on land surface temperature in mountainous city: a case study in central area of Chongqing City. Acta Ecologica Sinica, 2014，34(14):4017-4024. DOI: 10.5846/stxb201211271679.

14.Han Guifeng, Cai Zhi, Xie Yusi, Zeng Wei. Correlation between urban construction and urban heat island: A case study in Kaizhou District, Chongqing [J]. Journal of Civil and Environmental Engineering, 2016,38(05):138-147. DOI: 10.11835/j.issn.1674-4764.2016.05.018.

15.Han Xiuzhen, Li Sanmei, Dou Fangli. Study of obtaining high resolution near-surface atmosphere temperature by using the land surface temperature from meteorological satellite data[J]. Acta Meteorologica Sinica, 2012,70(05):1107-1118.

16.Hooker Josh, Duveiller Gregory, Cescatti Alessandro. A global dataset of air temperature derived from satellite remote sensing and weather stations [J]. Scientific Data, 2018,5. DOI:10.1038/sdata.2018.246.

17.Hu Zhaolin, Dai Hui, Hou Fei, Li Erzhu. Spatio-temporal change of urban-rural vegetation phenology and its response to land surface temperature in Northeast China. Acta Ecologica Sinica, 2020, 40(12):4137-4145. DOI:10.5846/stxb201812252806.

18.Jian Peng, Yuzhuo Dan, Ruilin Qiao, Yanxu Liu, Jianquan Dong, Jiansheng Wu. How to quantify the cooling effect of urban parks? Linking maximum and accumulation perspectives [J]. Remote Sensing of Environment, 2021,252:112-135. DOI.: 10.1016/j.rse.2020.112135.

19.Jin Lina, Li Xiongfei, Du Mengmeng, et al． Analysis on the Characteristics of Urban Heat Island in Xi’an Based on FY-3 Satellite Data [J]. Meteorological and Environmental Sciences, 2019,42( 4):74－82. DOI: 10.16765 /j.cnki.1673－7148. 2019.04.011

20.Jinlong Zang，Qiaozhen Guo，Xiaoxu Wu，Xiao Sang，Huanhuan Wu，Yue Qiao. Integrated evaluation on multi-scale land surface temperature grading and bio-temperature suitability—a case study in Tianjin China[J]. International Journal of Remote Sensing,2021,42(1).

21.Jiyao Zhao, Le Yu, Yidi Xu, Xuecao Li, Yuyu Zhou, Dailiang Peng, Han Liu, Xiaomeng Huang, Zheng Zhou, Dong Wang, Chao Ren, Peng Gong. Exploring difference in land surface temperature between the city centers and urban expansion areas of China’s major cities[J]. International Journal of Remote Sensing,2020,41(23). DOI:10.1080/01431161.2020.1797216.

22.Li Bin, Wang Huimin, Qin Mingzhou, Zhang Pengyan. Comparative study on the correlations between NDVI, NDMI and LST [J]. Progress in Geography, 2017,36(05):585-596. DOI: 10.18306/dlkxjz.2017.05.006.

23.Li Dan, Tian Peipei, Luo Hongying, Hu Tiesong, Dong Bin, Cui Yuanlai, Khan Shahbaz, Luo Yufeng. Impacts of land use and land cover changes on regional climate in the Lhasa River basin, Tibetan Plateau [J]. Science of The Total Environment, 2020,742. DOI: 10.1016/j.scitotenv.2020.140570.

24.Li Hui, Gu Ronghua, Zhu Yulin. Effect of Land Transfer on the Economic Radiation of Urban Agglomerations in the Middle Reaches of the Yangtze River and Formation Mechanism [J]. Resources and Environment in the Yangtze Basin, 2020,29(01):35-43. DOI: 10.11870/cjlyzyyhj202001004.

25.Li, Y., Schubert, S., Kropp, J. P., & Rybski, D. On the influence of density and morphology on the Urban Heat Island intensity[J]. Nature communications, 2020, 11(1):1-9. DOI: 10.1038/s41467-020-16461-9.

26.Liu Lu, Shen Guangrong, Wu Yu, ZhangZhou Yilin, Lu Shaoming. Characteristics of Land Use Transfer and Its Influence on Thermal Environment in County-level Urbanization [J]. Bulletin of Soil and Water Conservation, 2019,39(06):260-266. DOI:10.13961/j.cnki.stbctb.2019.06.038.

27.Liu Yansui. Research on the urban-rural integration and rural revitalization in the new era in China [J]. Acta Geographica Sinica, 2018,73(04):637-650. DOI:10.11821/dlxb201804004.

28.Liu Yaobin, Leng Qingsong. Urbanization, Population Agglomeration and Haze Changes: Based on Threshold Regression and Spatial Partition [J]. Ecological Economy, 2020,36(03):92-98.

29.Liu Ye, Xue Yongkang, Li Qian, Lettenmaier Dennis, Zhao Ping. Investigation of the Variability of Near‐Surface Temperature Anomaly and Its Causes Over the Tibetan Plateau[J]. Journal of Geophysical Research: Atmospheres,2020,125(19). DOI：10.1029/2020JD032800.

30.Liya Chao, Boyin Huang, Yang Yuanjian, Phil Jones, Jiayi Cheng, Yang, Qingxiang Li. A New Evaluation of the Role of Urbanization to Warming at Various Spatial Scales: Evidence From the Guangdong‐Hong Kong‐Macau Region, China[J]. Geophysical Research Letters,2020,47(20). DOI：

31.Mashhoodi Bardia, Stead Dominic, van Timmeren Arjan. Spatializing household energy consumption in the Netherlands: Socioeconomic, urban morphology, microclimate, land surface temperature and vegetation data [J]. Data in Brief, 2020,29. DOI:10.1016/j.dib.2020.105118.

32.Meng Qingyan, Sun Yunxiao, Zhang Jiahui, Chen Xu, Sun Zhenghui. Assessing Vertical Greenery Distribution and Spatial Allocation based on Multi-source Remote Sening:A Case Study of Székesfehérvár City, Hungary, [J]. Remote Sensing Technology and Application, 2018,33(02):370-376. DOI：10.11873/J.ISSN.1004-0323.2018.2.0370.

33.Milan Gavrilović,Milan Pjević,Mirko Borisov,Goran Marinković,Vladimir M. Petrović. Analysis of Climate Change in the Area of Vojvodina-Republic of Serbia and Possible Consequences[J]. Journal of Geographical Research,2020,02(02):11-19. DOI.10.30564/jgr.v2i2.952.

34.Mingxing Chen, Shasha Guo, Maogui Hu, Xiaoping Zhang. The spatiotemporal evolution of population exposure to PM 2.5 within the Beijing-Tianjin-Hebei urban agglomeration, China [J]. Journal of Cleaner Production, 2020. DOI.org/10.1016/j.jclepro.2020.121708.

35.National Bureau of Statistics http://data.stats.gov.cn

36.Oke T R. Canyon geometry and the nocturnal urban heat island: comparison of scale model and field observations [J]. International Journal of Climatology, 1981,1(3):237-254.

37.Qin Zhihao, Zhang Minghua, Arnon Karnieli, Pedro Berliner. Mono-window Algorithm for Retrieving Land Surface Temperature from Landsat TM6 data [J]. Acta Geographica Sinica, 2001(04):456-466. DOI:10.1142/S0252959901000401.

38.Qiu Lisha, He Yi, Zhang Lifeng, Wang Wenhui, Tang Yuanwei. Spatiotemporal variation characteristics and influence factors of MODIS LST in Qilian Mountains [J]. Arid Land Geography, 2020,43(03):721-737. DOI:10.12118/j.issn.1000-6060.2020.0319.

39.Ricardo Fonseca, María‐Paz Zorzano‐Mier, Armando Azua‐Bustos, Carlos González‐Silva, Javier Martín‐Torres. A surface temperature and moisture intercomparison study of the Weather Research and Forecasting model, in‐situ measurements and satellite observations over the Atacama Desert [J]. Quarterly Journal of the Royal Meteorological Society, 2019,145(722).

40.Scarano Mancini. Assessing the relationship between sky view factor and land surface temperature to the spatial resolution [J]. International Journal of Remote Sensing, 2017,38(23):6910-6929. DOI: 10.1080/01431161.2017.1368099

41.Shengjie Liu, Qian Shi. Local climate zone mapping as remote sensing scene classification using deep learning: A case study of metropolitan China [J]. ISPRS Journal of Photogrammetry and Remote Sensing, 2020,164:229-242 DOI.10.1016/j.isprsjprs.2020.04.008

42.Shi Xin, Zhou Maichun, Liu Zhenhua, Hu Yueming, Liu Yuan, Zhu Shuhua. Comparative Analysis on Three Land Surface Temperature Inversion Algorithm based on Landsat 8 Data over Sanheba Basin [J]. Remote Sensing Technology and Application, 2018,33(03):465-475. DOI:10.11873/j.isn.1004-0323.2018.3.0465.

43.Song Ting, Duan Zheng, Liu Junzhi, Shi Junzhe, Yan Fei, Sheng Shijie, Huang Jun, Wu Wei. Comparison of four algorithms to retrieve land surface temperature using Landsat 8 satellite [J]. Journal of Remote Sensing, 2015,19( 3) : 451-464. DOI: 10.11834 /jrs. 20154180.

44.Srivanit M, Kazunori H. The influence of urban morphology indicators on summer diurnal range of urban climate in Bangkok metropolitan area, Thailand [J]. International Journal of Civil & Environmental Engineering, 2011,11(5):34-46.

45.Stewart I D , Oke T R . Local Climate Zones for Urban Temperature Studies[J]. Bulletin of the American Meteorological Society, 2012, 93(12):1879-1900.DOI:10.1175/BAMS-D-11-00019.1

46.Stewart I D. A systematic review and scientific critique of methodology in modern urban heat island literature [J]. International Journal of Climatology, 2011,31(2):200-217. DOI.10.1002/joc.2141.

47.Tang Chengli, Chen Weiyang, Wu Jiamin Zhou Guohua, Wang Meixia, Guo Xiashuang. Spatial distribution and industrial agglomeration characteristics of development zones in the Yangtze River Economic Belt. Scientia Geographica Sinica, 2020,40(4):657-664. doi:10.13249/j.cnki.sgs.2020.04.018.

48.Wan K W, Tsang C L, Lam J C. Sensitivity analysis of building energy use in different climates [J]. IFAC Proceedings Volumes, 2010, 43(1):58-62.DOI. 10.3182/20100329-3-PT-3006.00013

49.Wang Hui, Wang Qi. The Pollution Emission and Urbanization in China：Based on Input-output Analysis [J]. Chinese Journal of Population Science, 2011(05):57-66+111-112.

50.Wu Qiong, Gong Jian, Yang Jianxin. Multi-scale Anaysis of Main Factors of Summer Thermal Field in Wuhan Based on POI Data [J]. Resources and Environment in the Yangtze Basin, 2020,29(01):200-210. DOI:10.11870/cjlyzyyhj202001018.

51.Xiaoni Wang, Catherine Prigent. Comparisons of Diurnal Variations of Land Surface Temperatures from Numerical Weather Prediction Analyses, Infrared Satellite Estimates and In Situ Measurements [J]. Remote Sensing, 2020,12(3).

52.Xue Yayong, Liang Haibin, Zhang Yuan, Wang Xiaofeng. Spatial and Temporal Variations of Land Surface Temperature of the Loess Plateau [J]. Earth and Environment, 2017,45(05):500-507. DOI: 10.14050 /j.cnki.1672-9250.2017.05.002

53.Yan Guanghua, Su Junru, Guan Dunyi. The Impact of Urban Architectural Vertical Characteristics on Urban Thermal Environment in Zhongshan District, Dalian [J]. Scientia Geographica Sinica, 2019,39(1): 125-130. DOI: 10.13249/j.cnki.sgs.2019.01.014.

54.Yang Chen, Zhan Q, Gao Sihang, Liu Huimin. Characterizing the spatial and temporal variation of the land surface temperature hotspots in Wuhan from a local scale[J]. Geo-spatial Information Science,2020,23(4). DOI: 10.1080/10095020.2020.1834882.

55.Yang, J., Sun, J., Ge, Q., & Li, X. (2017). Assessing the impacts of urbanization-associated green space on urban land surface temperature: A case study of Dalian, China[J]. Urban Forestry & Urban Greening, 2020,22: 1-10. DOI: 10.1016/j.ufug.2017.01.002.

56.Yanxu Liu, Jian Peng, Yanglin Wang. Efficiency of landscape metrics characterizing urban land surface temperature[J]. Landscape and Urban Planning,2018,180. DOI:10.1016/j.landurbplan.2018.08.006.

57.Yao Yuan, Chen Xi, Qian Jin. Research progress on the thermal environment of the urban surfaces．Acta Ecologica Sinica, 2018,38( 3):1134-1147. DOI: 10.5846 /stxb201611022233.

58.Zhang Hui, Yuan Fenghui, Wang anzhi, Guan Dexin, Dai Guanhua, Wu Jiabing. Variation characteristics of NDVI and its response to climatic change in the growing season of Changbai Mountain Nature Reserve during 2001 and 2018 [J]. Chinese Journal of Applied Ecology, 2020,31(04):1213-1222. DOI:10.13287/j.1001-9332.202004.020.

59.Zhang Yujia, Middel Ariane, Turner B L. Evaluating the effect of 3D urban form on neighborhood land surface temperature using Google Street View and geographically weighted regression [J]. Landscape Ecology, 2019. DOI: 10.1007/s10980-019-00794-y.

60.Zhao Zi-qi, Li Li-guang, Wang Hongbo, et al. Study on the relationships between land use and land surface temperature in Shenyang urban districts [J]. Journal of Meteorology and Environment, 2016, 32(6):102－108. DOI: 10.3969 /j.issn.1673-503X.2016.06.013.

61.Zhao, Z. Q., He, B. J., Li, L. G., Wang, H. B., & Darko, A. Profile and concentric zonal analysis of relationships between land use/land cover and land surface temperature: Case study of Shenyang, China[J]. Energy and Buildings, 2017, 155: 282-295. DOI: 10.1016/j.enbuild.2017.09.046.

62.Zheng, Z., Zhou, W., Yan, J., Qian, Y., Wang, J., & Li, W. The higher, the cooler? Effects of building height on land surface temperatures in residential areas of Beijing[J]. Physics and Chemistry of the Earth, Parts A/B/C, 2019,110:149-156. DOI: 10.1016/j.pce.2019.01.008.

63.Zhou Weiqi, Tian Yunyu. Effects of urban three-dimensional morphology on thermal environment: a review [J]. Acta Ecologica Sinica, 2020,40( 2):416-427. DOI: 10.5846 /stxb201902250353.

Amendmentl(L514 - L686 ):

Reference

1.Blessing Bolarinwa Fabeku, Ifeoluwa Adebowale Balogun, Suleiman Abdul-Azeez Adegboyega, et al. Spatio-Temporal Variability in Land Surface Temperature and Its Relationship with Vegetation Types over Ibadan, South-Western Nigeria [J]. Atmospheric and Climate Sciences, 2018,8(3):319-336.DOI:10.4236/acs.2018.83021.

2.Cao Jie, Zhou Weiqi, Zheng Zhong, Ren Tian, Wang Weimin. Within-city spatial and temporal heterogeneity of air temperature and its relationship with land surface temperature [J]. Landscape and Urban Planning, 2021,206. DOI: 10.1016/j.landurbplan.2020.103979.

3.Chao Liya, Huang Boyin , Yang Yuanjian, et al. A New Evaluation of the Role of Urbanization to Warming at Various Spatial Scales: Evidence From the Guangdong‐Hong Kong‐Macau Region, China[J]. Geophysical Research Letters,2020,47(20). DOI: 10.1029/2020GL089152.

4.Chen Dechao , Xu Xinliang , Sun Zongyao.et al., Assessment of Urban Heat Risk in Mountain Environments: A Case Study of Chongqing Metropolitan Area, China[J]. Sustainability,2019,12(1). DOI:10.3390/su12010309.

5.Chen Mingxing ,Guo Shasha , Hu Maogui, Zhang Xiaoping. The spatiotemporal evolution of population exposure to PM 2.5 within the Beijing-Tianjin-Hebei urban agglomeration, China [J]. Journal of Cleaner Production, 2020. DOI.org/10.1016/j.jclepro.2020.121708.

6.Cheng He., Liang Ma., Liguo Zhou.,et al. Exploring the mechanisms of heat wave vulnerability at the urban scale based on the application of big data and artificial societies [J]. Environment International, 2019,127. 573-583.DOI:10.1016/j.envint.2019.01.057.

7.Gu Hengyu, Qin Xiaoling, Shen Tiyan. Spatial variation of migrant population's return intention and its determinants in China's prefecture and provincial level cities [J]. Geographical Research, 2019,38(08):1877-1890. DOI: 10.11821/dlyj020180473.

8.Guo Andong.,Yang,Jun.,Sun Wei., et al. Impact of urban morphology and landscape characteristics on spatiotemporal heterogeneity of land surface temperature[J]. Sustainable Cities and Society,2020,63: 102443. DOI: 10.1016/j.scs.2020.102443.

9.Han Guifeng, Cai Zhi, Xie Yusi, Zeng Wei. Correlation between urban construction and urban heat island: A case study in Kaizhou District, Chongqing [J]. Journal of Civil and Environmental Engineering, 2016,38(05):138-147. DOI: 10.11835/j.issn.1674-4764.2016.05.018.

10.Han Guifeng, Ye Lin, Sun Zhongwei. Influence of aspect on land surface temperature in mountainous city: a case study in central area of Chongqing City[J]. Acta Ecologica Sinica, 2014，34(14):4017-4024. DOI: 10.5846/stxb201211271679.

11.He Baojie., Ding Lan., Deo Prasad., Enhancing urban ventilation performance through the development of precinct ventilation zones: A case study based on the Greater Sydney, Australia[J]. Sustainable Cities and Society,2019,47:DOI: 10.1016/j.scs.2019.101472.

12.He Baojie.,Zhao Ziqi.,Shen Lidu.,et al. An approach to examining performances of cool/hot sources in mitigating/enhancing land surface temperature under different temperature backgrounds based on landsat 8 image[J]. Sustainable Cities and Society,2019,44. DOI: 10.1016/j.scs.2018.10.049.

13.He, Bao-Jie., Wang, Junsong ., Liu, Huimin ., Ulpiani, Giulia. Localized synergies between heat waves and urban heat islands: Implications on human thermal comfort and urban heat management[J]. Environmental Research,2020,193. 110584. DOI:10.1016/j.envres.2020.110584. 

14.Hooker Josh, Duveiller Gregory, Cescatti Alessandro. A global dataset of air temperature derived from satellite remote sensing and weather stations [J]. Scientific Data, 2018,5. DOI:10.1038/sdata.2018.246.

15.Hu Zhaolin, Dai Hui, Hou Fei, Li Erzhu. Spatio-temporal change of urban-rural vegetation phenology and its response to land surface temperature in Northeast China. Acta Ecologica Sinica, 2020, 40(12):4137-4145. DOI:10.5846/stxb201812252806.

16.Li Bin, Wang Huimin, Qin Mingzhou, et al. Comparative study on the correlations between NDVI, NDMI and LST [J]. Progress in Geography, 2017,36(05):585-596. DOI: 10.18306/dlkxjz.2017.05.006.

17.Li Dan, Tian Peipei, Luo Hongying, et al. Impacts of land use and land cover changes on regional climate in the Lhasa River basin, Tibetan Plateau [J]. Science of The Total Environment, 2020,742. DOI: 10.1016/j.scitotenv.2020.140570.

18.Li, Yunfei, Schubert Sebastian, Kropp J. et al. On the influence of density and morphology on the Urban Heat Island intensity[J]. Nature communications, 2020, 11(1):1-9. DOI: 10.1038/s41467-020-16461-9.

19.Liu Lu., Shen Guangrong., Wu Yu., et al. Characteristics of Land Use Transfer and Its Influence on Thermal Environment in County-level Urbanization [J]. Bulletin of Soil and Water Conservation, 2019,39(06):260-266. DOI:10.13961/j.cnki.stbctb.2019.06.038.

20.Liu Shengjie, Shi Qian. Local climate zone mapping as remote sensing scene classification using deep learning: A case study of metropolitan China [J]. ISPRS Journal of Photogrammetry and Remote Sensing, 2020,164:229-242 DOI.10.1016/j.isprsjprs.2020.04.008

21.Liu Yanxu , Peng Jian , Wang Yanglin . Efficiency of landscape metrics characterizing urban land surface temperature[J]. Landscape and Urban Planning,2018,180. DOI:10.1016/j.landurbplan.2018.08.006.

22.Liu Yaobin, Leng Qingsong. Urbanization, Population Agglomeration and Haze Changes: Based on Threshold Regression and Spatial Partition [J]. Ecological Economy, 2020,36(03):92-98.

23.Luo Xue,Yang Jun,Sun Wei,et al. Suitability of human settlements in mountainous areas from the perspective of ventilation: A case study of the main urban area of Chongqing[J]. Journal of Cleaner Production,2021,310:DOI: 10.1016/J.JCLEPRO.2021.127467.

24.Mashhoodi Bardia, Stead Dominic, van Timmeren Arjan. Spatializing household energy consumption in the Netherlands: Socioeconomic, urban morphology, microclimate, land surface temperature and vegetation data [J]. Data in Brief, 2020,29. DOI:10.1016/j.dib.2020.105118.

25.Meng Qingyan, Sun Yunxiao, Zhang Jiahui,et al. Assessing Vertical Greenery Distribution and Spatial Allocation based on Multi-source Remote Sening:A Case Study of Székesfehérvár City, Hungary, [J]. Remote Sensing Technology and Application, 2018,33(02):370-376. DOI：10.11873/J.ISSN.1004-0323.2018.2.0370.

26.National Bureau of Statistics http://data.stats.gov.cn.

27.Oke T R. Canyon geometry and the nocturnal urban heat island: comparison of scale model and field observations [J]. International Journal of Climatology, 1981,1(3):237-254.

28.Peng Jian , Dan Yuzhuo , Qiao Ruilin ,et al. How to quantify the cooling effect of urban parks? Linking maximum and accumulation perspectives [J]. Remote Sensing of Environment, 2021,252:112-135. DOI.: 10.1016/j.rse.2020.112135.

29.Qin Zhihao, Zhang Minghua, Arnon Karnieli, Pedro Berliner. Mono-window Algorithm for Retrieving Land Surface Temperature from Landsat TM6 data [J]. Acta Geographica Sinica, 2001(04):456-466. DOI:10.1142/S0252959901000401.

30.Qiu Lisha, He Yi, Zhang Lifeng, Wang Wenhui, Tang Yuanwei. Spatiotemporal variation characteristics and influence factors of MODIS LST in Qilian Mountains [J]. Arid Land Geography, 2020,43(03):721-737. DOI:10.12118/j.issn.1000-6060.2020.0319.

31.Scarano Mancini. Assessing the relationship between sky view factor and land surface temperature to the spatial resolution [J]. International Journal of Remote Sensing, 2017,38(23):6910-6929. DOI: 10.1080/01431161.2017.1368099

32.Shi Xin, Zhou Maichun, Liu Zhenhua, et al. Comparative Analysis on Three Land Surface Temperature Inversion Algorithm based on Landsat 8 Data over Sanheba Basin [J]. Remote Sensing Technology and Application, 2018,33(03):465-475. DOI:10.11873/j.isn.1004-0323.2018.3.0465.

33.Song Ting, Duan Zheng, Liu Junzhi, et al. Comparison of four algorithms to retrieve land surface temperature using Landsat 8 satellite [J]. Journal of Remote Sensing, 2015,19( 3) : 451-464. DOI: 10.11834 /jrs. 20154180.

34.Stewart I D , Oke T R . Local Climate Zones for Urban Temperature Studies[J]. Bulletin of the American Meteorological Society, 2012, 93(12):1879-1900.DOI:10.1175/BAMS-D-11-00019.1

35.Stewart I D. A systematic review and scientific critique of methodology in modern urban heat island literature [J]. International Journal of Climatology, 2011,31(2):200-217. DOI.10.1002/joc.2141.

36.Stewart, I D.,Oke, T R. Local Climate Zone for Urban Temperatures Studies[J]. Bulletin of the American Meteorological Society,2012,93(12):DOI:10.1175/BAMS-D-11-00019.1.

37.Tang Chengli, Chen Weiyang, Wu Jiamin.et al. Spatial distribution and industrial agglomeration characteristics of development zones in the Yangtze River Economic Belt. Scientia Geographica Sinica, 2020,40(4):657-664. doi:10.13249/j.cnki.sgs.2020.04.018.

38.Wan K W, Tsang C L, Lam J C. Sensitivity analysis of building energy use in different climates [J]. IFAC Proceedings Volumes, 2010, 43(1):58-62.DOI. 10.3182/20100329-3-PT-3006.00013

39.Wang Hui, Wang Qi. The Pollution Emission and Urbanization in China：Based on Input-output Analysis [J]. Chinese Journal of Population Science, 2011(05):57-66+111-112.

40.Wang Xiaoni., Catherine Prigent. Comparisons of Diurnal Variations of Land Surface Temperatures from Numerical Weather Prediction Analyses, Infrared Satellite Estimates and In Situ Measurements [J]. Remote Sensing, 2020,12(3).DOI: 10.3390/rs12030583.

41.Wu Qiong, Gong Jian, Yang Jianxin. Multi-scale Anaysis of Main Factors of Summer Thermal Field in Wuhan Based on POI Data [J]. Resources and Environment in the Yangtze Basin, 2020,29(01):200-210. DOI:10.11870/cjlyzyyhj202001018.

42.Xue Yayong, Liang Haibin, Zhang Yuan, Wang Xiaofeng. Spatial and Temporal Variations of Land Surface Temperature of the Loess Plateau [J]. Earth and Environment, 2017,45(05):500-507. DOI: 10.14050 /j.cnki.1672-9250.2017.05.002.

43.Yan Guanghua, Su Junru, Guan Dunyi. The Impact of Urban Architectural Vertical Characteristics on Urban Thermal Environment in Zhongshan District, Dalian [J]. Scientia Geographica Sinica, 2019,39(1): 125-130. DOI: 10.13249/j.cnki.sgs.2019.01.014.

44.Yang Chen, Zhan Q, Gao Sihang, et al., Characterizing the spatial and temporal variation of the land surface temperature hotspots in Wuhan from a local scale[J]. Geo-spatial Information Science,2020,23(4). DOI: 10.1080/10095020.2020.1834882.

45.Yang Jun ,Jin Shanhe ,Xiao Xiangming et al. Local climate zone ventilation and urban land surface temperatures: Towards a performance-based and wind-sensitive planning proposal in megacities[J]. Sustainable Cities and Society,2019,47.DOI: 10.1016/j.scs.2019.101487.

46.Yang Jun,Wang Yichen,Xue Bing,et al. Contribution of urban ventilation to the thermal environment and urban energy demand: Different climate background perspectives[J]. Science of the Total Environment,2021,795:DOI: 10.1016/J.SCITOTENV.2021.148791.

47.Yang Jun.,Sun Jing.,Ge Quansheng.,et al. Assessing the impacts of urbanization-associated green space on urban land surface temperature: A case study of Dalian, China[J]. Urban Forestry & Urban Greening,2017,22:DOI: 10.1016/j.ufug.2017.01.002.

48.Zhang Hui, Yuan Fenghui, Wang anzhi, Guan Dexin, Dai Guanhua, Wu Jiabing. Variation characteristics of NDVI and its response to climatic change in the growing season of Changbai Mountain Nature Reserve during 2001 and 2018 [J]. Chinese Journal of Applied Ecology, 2020,31(04):1213-1222. DOI:10.13287/j.1001-9332.202004.020.

49.Zhang Yujia, Middel Ariane, Turner B L. Evaluating the effect of 3D urban form on neighborhood land surface temperature using Google Street View and geographically weighted regression [J]. Landscape Ecology, 2019. DOI: 10.1007/s10980-019-00794-y.

50.Zhao Jiyao ,Yu Le ,Xu Yidi, et al. Exploring difference in land surface temperature between the city centers and urban expansion areas of China’s major cities[J]. International Journal of Remote Sensing,2020,41(23). DOI:10.1080/01431161.2020.1797216.

51.Zhao Zi-qi, Li Li-guang, Wang Hongbo, et al. Study on the relationships between land use and land surface temperature in Shenyang urban districts [J]. Journal of Meteorology and Environment, 2016, 32(6):102－108. DOI: 10.3969 /j.issn.1673-503X.2016.06.013.

52.Zhao, Ziqi., He Baojie., Li Guangli., et al, An Profile and concentric zonal analysis of relationships between land use/land cover and land surface temperature: Case study of Shenyang, China[J]. Energy and Buildings, 2017, 155: 282-295. DOI: 10.1016/j.enbuild.2017.09.046.

53.Zheng Zhong.,Zhou Weiqi.,Yan Jingli.,et al. The higher, the cooler? Effects of building height on land surface temperatures in residential areas of Beijing[J]. Physics and Chemistry of the Earth,2019,110:149-156. DOI:10.1016/j.pce.2019.01.008.

54.Zhou Bin, Diego Rybski, Jürgen P. Kropp. The role of city size and urban form in the surface urban heat island [J]. Science Reports, 2017,(07):1-9. DOI.10.1038/s41598-017-04242-2.

55.Zhou Decheng , Xiao Jingfeng , Stefania Bonafoni, et al. Satellite Remote Sensing of Surface Urban Heat Islands: Progress, Challenges, and Perspectives[J]. Remote Sensing,2018,11(1). DOI:10.3390/rs11010048.

56.Zhou Weiqi, Tian Yunyu. Effects of urban three-dimensional morphology on thermal environment: a review [J]. Acta Ecologica Sinica, 2020,40( 2):416-427. DOI: 10.5846 /stxb201902250353.

Comment 3: Moreover， authors are required to provide a document of 'responses to reviewers' in which authors' responses are provided point-by-point.

Answers to comment 3: 

Thank you very much for referees’ reports. I seriously thought about the reviewer's opinion and carefully answered the question. After the revision, I have been more cautious about the content that reviewers focused on. Named as the response to the reviewer, the document, as a separate file, is attached to the response to the response to the manuscript. 

Comment 4:Therefore, please reply to me in the next round based on the document of 'responses to reviewers'. Both the first-round and second-round responses should be provided.

Answers to comment 4:

Thank you very much for referees’ reports. I seriously thought about the reviewer's opinion and carefully answered the question. After the revision, I am willing to provide additional documents for the first and second rounds of revision responses. Nominated as the response to the reviewer, the files mentioned above are attached to the response to the manuscript. 

Comment 5:Section 6 should be shortened to provide key information.

Answers to comment 5:

Thank you very much for referees’ reports. After intense internal brain-storming, we seriously reviewed the sixth section. Strongly and humbly, we agreed with and adopted the comments of the reviewers. Based on the above internal-external joint contributions, we have made the following changes to the parts that reviewers focused on. 

On the one hand, based on the selection of impact factors, two-dimensional plane combined with three-dimensional multi-dimensional research perspectives, and research methods, we have re-written the advantage section. By comparing the latest research progress, the manuscript expanded the research perspective of LST, starting from a two-dimensional plane and extending it to a three-dimensional space, further quantifying the impact of different spatial organization types on LST. It is necessary to mention that the manuscript in an earlier period combined MGWR model with LST research , which further proved the correlation of urban spatial structure on LST. 

On the other hand, in the limitation section, instead of the previous expressions that may have caused readers' misunderstanding, we have rewritten some expressions that are more likely to arouse readers' discussion. As MGWR model is more widely used in thermal environment research, we look forward to receiving more feedback. In addition to calling on people to pay more attention to the relevance of their own development and environmental changes, we are more inclined to seek solutions, whether it is economic policies or law regulations. Once again, I sincerely thank the referees for their careful reports and predecessors for their painstaking research.

Originall(L425 - L503 ):

6. Advantages and Limitations

The conclusions of this research and existing research results are mutually confirmed (Han et al., 2016; Yan et al., 2019). In plain areas with low altitude and gentle slopes, the development of human activities is less difficult, human activities are frequent, and construction land is concentrated. Areas with concentrated secondary and tertiary industries are more likely to have higher LST, whereas areas with mountains and agricultural land are likely to have lower LST due to several factors such as altitude and vegetation. It is noteworthy that, unlike the findings of previous studies, an obvious positive correlation was found between urban BH, BD, and LST. However, the LST of high-rise and super high-rise building areas was found to be lower than that of mid-rise building areas. This could be explained by the expansion of shadow areas generated by super high-rise buildings; similar phenomena were also observed between medium-density building areas and high-density building areas (Meng et al., 2018; Peng et al., 2021).

A city is a complex system where residents provide certain infrastructure and social relations. The urban thermal environment is the result of a combination of human activities and other factors under certain natural conditions (Zhao et al.,2020; Chen et al.,2019). Due to the problem of data acquisition, the research on the relationship between surface temperature and its influence at home and abroad mostly adopts top-down methods, to a certain extent Ignore the influence of human activities and urban form. 

Based on the existing research, this paper determines the impact of different human activities on the urban thermal environment, and further proves that urban green space can help alleviate the urban heat island effect. (Zhou et al.,2018; Zhao et al.,2020). The process of urbanization has led to overpopulation and excessive industrial concentration, causes a change in the nature of heat exchange at the bottom, and aggravating the formation and development of the urban heat island effect, which requires the attention of urban planning agencies. In addition, based on the results of this article, strategies to reduce heat stress by addressing the urban heat island effect, (e.g., control the scale of urban built-up areas, optimize urban spatial structure, increase urban green areas, alleviate urban population concentration and other measures). We should also cooperate with commercial real estate developers to control the height and density of new buildings, optimize the design of future urban parks, increase the construction of urban ventilation corridors and green spaces, and further alleviate the urban heat island effect (Liu et al.,2020).

6.1 Advantages

Based on research of urban thermal environment effect for a single data source influencing factors, and the problem of analysis method of single with higher level of urbanization in Nanjing as the research object, combined with digital elevation data urban building land use type data and POI data set multi-source spatial information data, comprehensive utilization of quantitative inversion and MGWR analysis methods, such as, on the basis of comprehensive study and the urban thermal environment effect and influence factors, further enrich the research of urban thermal environment research perspective Studies have proved that Geography Weighted Regression(GWR) has been broadly used in various fields to model spatially non-stationary relationships(Liu et al.,2018;Yang et al.,2020).Multi-scale Geographically Weighted Regression(MGWR) is a recent advancement to the classic GWR model. Compared with the traditional GWR model, The MGWR model has advantages in acquiring the ability of different scales (Jin et al.,2021). The MGWR model can effectively analyze the multi-scale relationship between the urban thermal environment and its influencing factors, and has a positive effect on urban dynamic development and urban thermal environment management.

6.2 Limitations

This study integrated POI data, building data, urban road data sets and other data to analyze the factors affecting the urban thermal environment. However, it should also be noted that POI data represents the geographic information and utilization characteristics of various facilities, and can only represent a certain Whether the utilization characteristics of each time node can represent the historical utilization characteristics and intensity still needs to be studied. Secondly, as for the classification of building height in Nanjing, this study is based on the established unified standard. However, if we can carry out extensive discussion based on the actual situation of the study area, and finally determine the most reasonable height classification, the research will have more practical value. Thirdly, existing researches mostly focus on the status quo of urban thermal environment, and there is still a lack of in-depth research on the prediction and analysis of urban thermal environment evolution trends, the construction of a heat island effect warning mechanism, and urban space optimization strategies.

The MGWR model facilitated multi-factor analysis of LST. However, due to issues such as data availability and collinearity, the application of the MGWR model has certain limitations. The urban landscape is a complex dynamic system composed of infrastructure, human activities, and social connections. Changes in urban surface temperature need to be observed from a more micro perspective (Cao et al., 2021; Li et al., 2020). Urban ground monitoring data have not been fully disclosed, which limits the study. In addition, street view data were used in the study of the urban thermal environment, and the number of street scenes in this area requires further investigation (Zhang et al., 2019). In the future, the interaction between different influencing factors should be considered, and the influencing factors of LST should be analyzed in more detail to provide a more comprehensive perspective for urban or regional environmental governance and planning.

Amendmentl(L426 - L490 ):

6. Advantages and Limitations

The conclusions of this research and existing research results are mutually confirmed (Han et al., 2014; Yan et al., 2019;Yang et al.,2020). In plain areas with low altitude and gentle slopes, the development of human activities is less difficult, human activities are frequent, and construction land is concentrated. Areas with concentrated secondary and tertiary industries are more likely to have higher LST, whereas areas with mountains and agricultural land are likely to have lower LST due to several factors such as altitude and vegetation(Luo et al.,2021;Peng et al.,2021). It is noteworthy that, unlike the findings of previous studies, an obvious positive correlation was found between urban BH, BD, and LST. However, the LST of high-rise and super high-rise building areas was found to be lower than that of mid-rise building areas. This could be explained by the expansion of shadow areas generated by super high-rise buildings; similar phenomena were also observed between medium-density building areas and high-density building areas (Meng et al., 2018; Yang et al., 2021).

Based on the existing research, this paper determines the impact of different human activities on the urban thermal environment, and further proves that urban green space can help alleviate the urban heat island effect. (Yang et al.,2017;Zhao et al.,2020). The process of urbanization has led to overpopulation and excessive industrial concentration, causes a change in the nature of heat exchange at the bottom, and aggravating the formation and development of the urban heat island effect, which requires the attention of urban planning agencies. In addition, based on the results of this article, strategies to reduce heat stress by addressing the urban heat island effect, (e.g., control the scale of urban built-up areas, optimize urban spatial structure, increase urban green areas, alleviate urban population concentration and other measures). We should also cooperate with commercial real estate developers to control the height and density of new buildings, optimize the design of future urban parks, increase the construction of urban ventilation corridors and green spaces, and further alleviate the urban heat island effect (Liu et al.,2020).

6.1 Advantages

This study integrates POI data, building data, urban road data sets and other data to analyze the factors that affect the urban thermal environment. First, in order to assess the human activities that may be responsible for the model described here, we compare land use data with population density data, etc., and conclude that the POI data represents the geographic information and utilization characteristics of various facilities. Secondly, for the classification of building height in Nanjing, after many field investigations and analysis of historical remote sensing images, a more reasonable density classification (limited to 40%) is finally determined. This research will have greater practical value. Third, this study uses a multi-dimensional perspective (two-dimensional and three-dimensional structure) to study the current status of Nanjing's thermal environment, and explores the current status of Nanjing's thermal environment from a more specific plane dimension and a deeper perspective. In view of the multi-dimensional perspective of this study, MGWR is used instead of GWR to explore non-stationary relationships in the modeling space (Liu et al., 2018; Yang et al., 2020).

Multi-scale Geographically Weighted Regression(MGWR) is a recent advancement to the classic GWR model. Compared with the traditional GWR model, The MGWR model has advantages in acquiring the ability of different scales (Jin et al.,2021). The MGWR model can effectively analyze the multi-scale relationship between the urban thermal environment and its influencing factors, and has a positive effect on urban dynamic development and urban thermal environment management.

6.2 Limitations

The MGWR model facilitated multi-factor analysis of LST. However, due to issues such as data availability and collinearity, the application of the MGWR model has certain limitations. The urban landscape is a complex dynamic system composed of infrastructure, human activities, and social connections. Changes in urban surface temperature need to be observed from a more micro perspective (Cao et al., 2021; Li et al., 2020). Urban ground monitoring data have not been fully disclosed, which limits the study. In addition, street view data were used in the study of the urban thermal environment, and the number of street scenes in this area requires further investigation (Zhang et al., 2019). In the future, the interaction between different influencing factors should be considered, and the influencing factors of LST should be analyzed in more detail to provide a more comprehensive perspective for urban or regional environmental governance and planning.

---

## [Decision Letter · Decision Letter 2]

5 Nov 2021

Relationship between urban morphology and land surface temperature – a case study of Nanjing City

PONE-D-21-03097R2

Dear Dr. shusheng,

We’re pleased to inform you that your manuscript has been judged scientifically suitable for publication and will be formally accepted for publication once it meets all outstanding technical requirements.

Kind regards,

Jun Yang

Academic Editor

PLOS ONE

Additional Editor Comments (optional):

Accept

Reviewers' comments:

Reviewer's Responses to Questions

**Comments to the Author**

1. If the authors have adequately addressed your comments raised in a previous round of review and you feel that this manuscript is now acceptable for publication, you may indicate that here to bypass the “Comments to the Author” section, enter your conflict of interest statement in the “Confidential to Editor” section, and submit your "Accept" recommendation.

Reviewer #1: All comments have been addressed

Reviewer #2: All comments have been addressed

2. Is the manuscript technically sound, and do the data support the conclusions?

Reviewer #1: Yes

Reviewer #2: Yes

3. Has the statistical analysis been performed appropriately and rigorously? 

Reviewer #1: Yes

Reviewer #2: Yes

4. Have the authors made all data underlying the findings in their manuscript fully available?

Reviewer #1: Yes

Reviewer #2: Yes

5. Is the manuscript presented in an intelligible fashion and written in standard English?

Reviewer #1: Yes

Reviewer #2: Yes

6. Review Comments to the Author

Reviewer #1: This paper is now well written. The English is good and the results are robust. Therefore, it is now acceptable.

Reviewer #2: As I was satisfied with the authors' response to my comments in their first revision, I will defer to the other reviewers and editor on any remaining issues.

7. PLOS authors have the option to publish the peer review history of their article (what does this mean?). If published, this will include your full peer review and any attached files.

Reviewer #1: No

Reviewer #2: No

---

## [Editor Report · Acceptance letter]

21 Jan 2022

PONE-D-21-03097R2 

Relationship between urban morphology and land surface temperature –– a case study of Nanjing City 

Dear Dr. Yin:

I'm pleased to inform you that your manuscript has been deemed suitable for publication in PLOS ONE. Congratulations! Your manuscript is now with our production department. 

Kind regards, 

on behalf of

Dr. Jun Yang 

Academic Editor

PLOS ONE